# Differential V2-directed antibody responses in non-human primates infected with SHIVs or immunized with diverse HIV vaccines

Svenja Weiss [1], Vincenza Itri[1], Ruimin Pan[2], Xunqing Jiang[2], Christina C. Luo[2], Lynn Morris [3,4], Delphine C. Malherbe [5,9], Philip Barnette[5], Jeff Alexander [6,10], Xiang-Peng Kong[2], Nancy L. Haigwood [5], Ann J. Hessell [5], Ralf Duerr [7] & Susan Zolla-Pazner [1,8 ✉]

V2p and V2i antibodies (Abs) that are specific for epitopes in the V1V2 region of the HIV gp120 envelope (Env) do not effectively neutralize HIV but mediate Fc-dependent anti-viral activities that have been correlated with protection from, or control of HIV, SIV and SHIV infections. Here, we describe a novel molecular toolbox that allows the discrimination of antigenically and functionally distinct polyclonal V2 Ab responses. We identify different patterns of V2 Ab induction by SHIV infection and three separate vaccine regimens that aid in fine-tuning an optimized immunization protocol for inducing V2p and V2i Abs. We observe no, or weak and sporadic V2p and V2i Abs in non-vaccinated SHIV-infected NHPs, but strong V2p and/or V2i Ab responses after immunization with a V2-targeting vaccine protocol. The V2-focused vaccination is superior to both natural infection and to immunization with whole Env constructs for inducing functional V2p- and V2i-specific responses. Strikingly, levels of V2-directed Abs correlate inversely with Abs specific for peptides of V3 and C5. These data demonstrate that a V1V2-targeting vaccine has advantages over the imprecise targeting of SIV/SHIV infections and of whole Env-based immunization regimens for inducing a more focused functional V2p- and V2i-specific Ab response.

[1] Department of Medicine, Division of Infectious Diseases, Icahn School of Medicine at Mount Sinai, New York, NY, USA. [2] Department of Biochemistry and Molecular Pharmacology, New York University School of Medicine, New York, NY, USA. [3] National Institute for Communicable Diseases, National Health Laboratory Service, Sandringham, Johannesburg, South Africa. [4] MRC Antibody Research Unit, University of the Witwatersrand, Johannesburg and Center for the AIDS Program of Research in South Africa, Johannesburg, South Africa. [5] Oregon National Primate Research Center, Oregon Health and Science University, Beaverton, OR, USA. [6] PaxVax Corporation, Redwood City, CA, USA. [7] Department of Microbiology, New York University School of Medicine, New York, NY, USA. [8] Department of Microbiology, Icahn School of Medicine, New York, NY, USA. [9] Present address: University of Texas Medical Branch, Department of Pathology, Galveston National Laboratory, Galveston, TX, USA. [10] Present address: JL Alexander Research and Development Consulting LLC, San Diego, CA, USA. ✉email: Susan.Zolla-Pazner@mssm.edu

RV144 is the only Phase 2b/3 clinical vaccine trial to date to demonstrate modest but significant efficacy in preventing HIV infection[1]. Subsequent studies of specimens from RV144 volunteers indicated that the only primary, independent correlate of reduced risk (CoR) was a robust level of non-neutralizing Abs binding to a recombinant protein containing the first and second variable regions (V1V2) of gp120, a domain in the envelope (Env) glycoprotein of HIV-1[2–4]. Later experiments demonstrated a significant inverse CoR with binding to V2 peptides[5]. These studies generated the hypothesis that Abs directed against V1V2 contributed to the reduced incidence in HIV infections in vaccinees[1–4,6].

The findings from RV144 continue to be debated[7–9], especially following the failure of the recent HVTN 702 vaccine trial in South Africa to demonstrate efficacy[10], although the HVTN 702 immunization regimen differed in several aspects from RV144[11]. Nonetheless, the RV144 CoR with V1V2 Abs has been buttressed by similar conclusions emanating from several studies of immunized NHPs in which protection, control, and/or delayed infection with SIV or SHIV were correlated with strong Ab responses to the V1V2 domain of gp120 (reviewed in Zolla-Pazner et al.[8]). These findings have given rise to the hypothesis that efficient induction of V2-specific Abs would aid in preventing HIV infection.

Three V1V2 domains form the apex of the Env trimer, which upon Env binding to CD4, undergoes extreme conformational changes allowing access to the coreceptor binding site[12]; this is a requisite process for the initiation of infection. The structurally stabilized V1V2 domain forms a five-stranded β-barrel[13,14], in the prefusion trimer, however, its C-strand (V2C) exists in both α-helical and β-strand configurations depending on the V1V2 sequence, its molecular environment, and structural constraints[13–20].

Many human anti-V1V2 monoclonal Abs (mAbs) have been isolated and the epitopes recognized by these mAbs have been classified into four families: V2i, V2p, V2q, and V2qt[8,15,21,22]. V2i-specific mAbs, including mAbs 830A and 2158, recognize V2 when its V2C region is in a β-strand configuration; the epitope region of these V2i mAbs is discontinuous, highly conformational, and overlaps the α4β7 integrin-binding motif[13,21]. V2p mAbs, including mAbs CH58 and CAP228-16H, were isolated from a recipient of the RV144 vaccine regimen and an infected individual, respectively, and target the V2C strand region as an α-helix and extended coil[16,19,23]. The epitope region recognized by V2q mAbs, such as mAbs PG9 and PG16, includes two N-linked glycans, e.g., N156 and N160, and consists of the V2C in its β-strand configuration[18,24–27], while the epitope region recognized by V2qt mAbs, such as PGT145 and PGDM1400, is located at the axial center of the Env trimer[28,29].

V2q and V2qt mAbs display broad and potent virus-neutralizing activities[29–33] as well as other Ab effector functions (reviewed in Duerr and Gorny[7]). As yet, no vaccine has induced broad and potent V2q or V2qt Abs in humans or any animal models. While V2p and V2i mAbs are poor neutralizers, they are effective at mediating various Fc-dependent anti-viral activities[20,34–37]. An ever-increasing number of studies indicates that Abs with anti-viral functions mediated by the Fc-fragment of Abs are effective against various viral infections, including HIV, SIV, SHIV, influenza, SARS-CoV-2, and herpes[38–52].

Several vaccine regimens have been shown to induce Abs reactive with V2 peptides or V1V2-scaffold protein in rabbits, non-human primates (NHPs)[22,53,54] and humans[2–4]. However, little is known about the fine specificities of the vaccine-induced polyclonal V2 Ab responses, the relative levels of V2p vs. V2i Abs in serum, or how robust V2-specific Ab responses can best be induced. Furthermore, no study has directly investigated V1V2 responses in SHIV-infected vs. vaccinated NHPs. Therefore, we undertook studies to elucidate the fine specificity and function of V2p and V2i Abs in SHIV-infected and vaccinated NHPs with the goal of providing valuable insights for the design of vaccines for both prevention of infection and therapeutic control of viral replication.

Here, we systematically probed the nature of the V1V2 Ab response of 20 SHIV-infected and 25 immunized rhesus macaques using one of three different vaccine regimens. We used a newly developed panel of antigens (Ags) that presents V2 or V1V2 in various molecular settings in which V2C assumes α-helical and/or β-stranded configurations[13,15–18,55] that react with V2p and/or V2i mAbs, respectively. The results show that in NHPs infected for 3–4 months with Tier 1 or Tier 2 SHIVs, essentially no V2p or V2i Abs were induced. Immunizations with env/gag DNA or a recombinant adenovirus vector along with gp120 or gp140 induced V2p Abs. In contrast, animals receiving env DNA and V1V2-scaffold protein immunogens mounted targeted functional Ab responses of both the V2p and V2i types, underscoring the influence of vaccine composition for developing strength, specificity, and functionality of V2 Ab responses.

## Results

**V2-specific mAbs distinguish between HIV-1 Env peptides, proteins, and V1V2-scaffold proteins.** To assess their antigenic nature, three gp120 proteins, five cyclic V2 (cV2) peptides, three linear V2 peptides[56], and eleven V1V2-fusion proteins were tested for reactivity with five V2p-, four V2i-, one V2q-, and one V2qt-specific mAbs. Anti-V3 mAb 447-52D and linear V3 and C5 peptides were used as controls. The sequences of V2p epitopes of several of the reagents used in this study are shown in Supplementary Table S1. The results, shown in Fig. 1a and Supplementary Fig. S2, from multiplex bead Ab binding assays using these reagents, are represented as area-under-the-curve (AUC) values that were derived from titration curves (Supplementary Fig. S1). The data indicate that V2p and V2i mAbs display extensive cross-reactivity and different patterns of binding reactivity: V2p mAbs recognize cV2 peptides from many clades but are non-reactive with three of four V1V2-1FD6 molecules. For example, strong cross-reactivity is seen in the data in Fig. 1a and Supplementary Fig. S2 by mAb V2p CAP228-16H with reagents from three clade C strains (ZM53, 1086, 97ZA012), two clade A strains (MG505 and 92RW020), two clade AE strains (92TH023 and A244) and one clade B strain (CaseA2) although, as shown in Supplementary Table S1, strains C/1086 and A/MG505 differ in sequence by 32% (6 of 19 positions in V2) and yet have similar and strong reactivity with CAP228-16H. These data support crystallographic data indicating that V2p mAbs recognize the V2C strand in its alpha-helical configuration[19,20]. Two of these V2p mAbs (mAbs CH58 and CH59) were derived from individuals immunized with the RV144 vaccine regimen consisting of immunogens from clades B and AE[19], while the other three V2p mAbs (CAP228-19F, −3D.1, and −16H) were derived from a clade C-infected individual[20]. Notably, these five mAbs are cross-reactive with antigens derived from clades A, C, and AE, but not with clade B.

In contrast, V2i mAbs, all of which were derived from clade B-infected individuals[57–60], react with V1V2-1FD6 Ags, but bind poorly, if at all, with cV2 peptides. They are highly cross-clade reactive with Ags from clades A, B, C, and AE. This broad cross-reactivity is consistent with their recognition of the conserved β-barrel conformation of the V1V2 domain[13,15,21]. Statistical analyses comparing binding of V2p and V2i mAbs to cV2 peptides and V1V2-1FD6 proteins reflect these differential

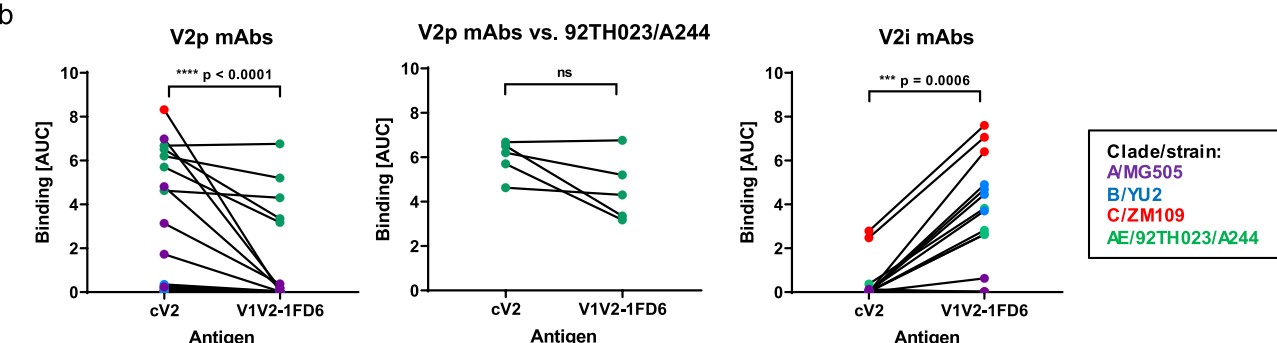

| | mAb | CH58 | CH59 | CAP228 19F | CAP228 3D.1 | CAP228 16H | 830A | 697-30D | 2158 | 1361 | PG9 | PGT145 | 447-52D |
|---|---|---|---|---|---|---|---|---|---|---|---|---|---|
| | Specificity | | V2p | | | | | V2i | | | V2q | V2qt | V3 |
| **Antigen** | | | | | | | | | | | | | |
| gp120 | gp120$_{C/ZM53}$ | 3.18 | 0.00 | 7.22 | 6.46 | 6.94 | 6.66 | 5.06 | 5.86 | 6.19 | 2.08 | 0.00 | 6.67 |
| | gp120$_{B/YU2}$ | 0.01 | 0.01 | 0.02 | 0.00 | 0.02 | 5.49 | 3.55 | 4.06 | 3.93 | 0.04 | 0.00 | 9.82 |
| | gp120$_{A/MG505}$ | 0.45 | 0.01 | 3.95 | 3.11 | 3.64 | 3.52 | 0.67 | 1.47 | 1.58 | 0.11 | 0.00 | 7.62 |
| Peptides | V3$_{C/consC}$ | 0.12 | 0.03 | 0.02 | 0.02 | 0.07 | 0.07 | 0.15 | 0.04 | 0.04 | 0.00 | 0.00 | 8.90 |
| | C5$_{C/ZM109}$ | 0.13 | 0.04 | 0.07 | 0.04 | 0.15 | 0.03 | 0.02 | 0.02 | 0.03 | 0.00 | 0.00 | 0.01 |
| | cV2$_{AE/92TH023}$ | 5.70 | 6.51 | 6.20 | 4.63 | 6.68 | 0.37 | 0.04 | 0.01 | 0.01 | 0.00 | 0.00 | 0.01 |
| | cV2$_{C/ZM109}$ | 0.09 | 8.32 | 0.03 | 0.01 | 0.06 | 2.48 | 0.06 | 0.00 | 2.79 | 0.00 | 0.00 | 0.03 |
| | cV2$_{C/1086}$ | 6.69 | 7.79 | 7.89 | 6.50 | 7.66 | 1.23 | 0.63 | 0.10 | 0.03 | 0.00 | 0.00 | 0.03 |
| | cV2$_{B/YU2}$ | 0.26 | 0.35 | 0.14 | 0.05 | 0.18 | 0.01 | 0.07 | 0.00 | 0.00 | 0.00 | 0.00 | 0.03 |
| | cV2$_{A/MG505}$ | 1.73 | 0.24 | 4.81 | 3.14 | 6.99 | 0.00 | 0.13 | 0.00 | 0.00 | 0.00 | 0.00 | 0.02 |
| | V2(AA166-185)$_{AE/A244}$ | 9.21 | 9.73 | 10.26 | 9.64 | 10.73 | 0.01 | 0.10 | 0.00 | 0.00 | 0.00 | 0.00 | 0.04 |
| | V2(AA172-191)$_{AE/A244}$ | 9.11 | 9.29 | 10.31 | 8.00 | 11.23 | 0.01 | 0.04 | 0.00 | 0.00 | 0.00 | 0.00 | 0.03 |
| | V2(AA190-209)$_{AE/A244}$ | 0.57 | 0.90 | 0.19 | 0.05 | 0.22 | 0.01 | 0.05 | 0.00 | 0.00 | 0.00 | 0.00 | 0.03 |
| V1V2-tags | V1V2$_{AE/A244}$-tags | 4.13 | 5.45 | 3.97 | 4.21 | 4.00 | 3.32 | 0.40 | 2.32 | 1.72 | 0.03 | 0.00 | 0.00 |
| | V1V2$_{C/1086}$-tags | 6.05 | 7.29 | 7.18 | 5.88 | 7.02 | 0.83 | 0.01 | 0.06 | 0.79 | 0.00 | 0.00 | 0.00 |
| | V1V2$_{B/CaseA2}$-tags | 0.01 | 0.00 | 0.23 | 0.01 | 0.21 | 4.72 | 3.92 | 4.37 | 4.59 | 0.00 | 0.00 | 0.00 |
| V1V2-gp70s | V1V2$_{C/97ZA012}$-gp70 | 0.01 | 0.69 | 7.63 | 1.52 | 7.07 | 5.87 | 5.82 | 6.31 | 6.57 | 0.00 | 0.00 | 0.00 |
| | V1V2$_{B/HxB2}$-gp70 | 0.01 | 0.00 | 0.78 | 0.01 | 0.57 | 6.34 | 5.61 | 6.08 | 7.45 | 0.00 | 0.00 | 0.00 |
| | V1V2$_{B/CaseA2}$-gp70 | 1.04 | 0.00 | 3.51 | 1.80 | 3.74 | 5.27 | 4.35 | 5.43 | 5.58 | 0.00 | 0.00 | 0.00 |
| | V1V2$_{A/92RW020}$-gp70 | 8.87 | 3.30 | 9.01 | 9.23 | 9.23 | 7.78 | 0.05 | 4.68 | 6.53 | 0.00 | 0.00 | 0.00 |
| V1V2-1FD6s | V1V2$_{AE/A244}$-1FD6 | 3.18 | 3.35 | 5.20 | 4.30 | 6.76 | 3.82 | 2.64 | 2.82 | 2.63 | 0.32 | 0.00 | 0.01 |
| | V1V2$_{C/ZM109}$-1FD6 | 0.01 | 0.01 | 0.00 | 0.01 | 0.01 | 7.06 | 0.03 | 6.40 | 7.60 | 0.37 | 0.00 | 0.00 |
| | V1V2$_{B/YU2}$-1FD6 | 0.04 | 0.04 | 0.03 | 0.03 | 0.03 | 4.90 | 3.70 | 4.45 | 4.69 | 0.00 | 0.03 | 0.03 |
| | V1V2$_{A/MG505}$-1FD6 | 0.03 | 0.00 | 0.18 | 0.38 | 0.19 | 0.63 | 0.00 | 0.00 | 0.01 | 0.00 | 0.00 | 0.00 |

**Clade:**
A
B
C
E

**Binding in AUC**
15-10 Very strong
10-5 strong
5-1 moderate
1-0.5 weak
0.5-0 no

**Fig. 1 Reactivity of HIV-1 Env antigens with V2 monoclonal antibodies. a** A multiplex bead binding assay was used to determine the levels of reactivity of mAbs specific for different V2 epitopes with various HIV-1 antigens. The clade from which each antigen was derived is indicated by the subscripted letter and by color coding. As a control, V3 mAb 447-52D was used. Irrelevant mAbs and PBS were used as negative controls in each experiment (not shown). Data are shown as AUC values generated from titration curves for each mAb, provided in Supplementary Fig. S1. Strength of binding is color-coded as per the spectrum shown in the figure. Experiments were performed at least twice. **b** Statistical analyses comparing binding to cV2 peptides and V1V2-1FD6 proteins by V2p mAbs (CH58, CH59, CAP228_19F, CAP228_3D.1, and CAP228_16H) ($p < 0.0001$) (left graph) and V2i mAbs (830 A, 697-30D 2158, 1361) ($p = 0.0006$) (right graph) are shown along with a comparison of V2p mAbs (CH58, CH59, CAP228_19F, CAP228_3D.1, and CAP228_16H) vs cV2 peptides and V1V2-1FD6 proteins derived from HIV$_{AE/92TH023}$ and HIV$_{AE/A244}$ ($p = 0.125$) (center graph). Color of symbols show the clade/strain of the antigen used as per those used in Fig. 1a and Supplementary S2. Lines connect pairs of a given mAb with reagents from a given strain. For example, the connected red symbols in the left panel of Fig. 1b represent the reactivity of V2p mAb CH59 with cV2$_{C/ZM109}$ and V1V2$_{C/ZM109}$-1FD6. Statistical analyses were performed with the two-tailed Wilcoxon matched-pairs signed-rank test and are shown for: all HIV-1 strains used in Figure1b (left and right graphs) or for V2p mAbs reacting to the HIV-1 strain A244 (middle graph). Source data are provided as a Source Data file. AA amino acid, Ab antibody, Ags antigens, AUC area-under-the-curve, consC consensus C, cV2 cyclic V2, Env envelope, mAb monoclonal antibody.

activities. Binding of V2p mAbs to cV2 peptides was significantly stronger than binding to V1V2-1FD6 ($p < 0.0001$), while binding of V2i mAbs was significantly stronger to V1V2-1FD6 than binding to cV2 peptides ($p = 0.0006$; Fig. 1b).

Interesting exceptions were noted with particular antigen/mAb pairs: there was no significant difference in binding of either V2p or V2i mAbs to clade AE-based cV2 peptides or V1V2-1FD6 proteins suggesting that the V2C region of strain AE/92TH023 transitions between an α-helix and a β-strand (Figs. 1b and 2b). In contrast, the V2C region exists solely as an α-helix/random coil in cV2$_{C/1086}$ and cV2$_{A/MG505}$ (Fig. 2b). Another exception is V2p mAb CAP228-16H which can react with three V1V2-1FD6

scaffold proteins from clade C[16] indicating that V2C regions in some clade C strains can form α-helices and/or β-strands.

**Identification of V1V2 antigens for defining the reactivities of V2p and V2i Abs.** To examine the relative presence of α-helix/random coil and/or β-strand configurations in V1V2- or V2-peptides in solution, CD spectra were generated and fitted using the CD-FIT program (http://www.ruppweb.org/Xray/comp/cdfit.htm; Fig. 2a, b). Analyses of various cV2s from clades A, B, C, and AE identified 81–100% of the secondary structures as α-helix/random coils, and 0–19% as β-strands. In contrast, when

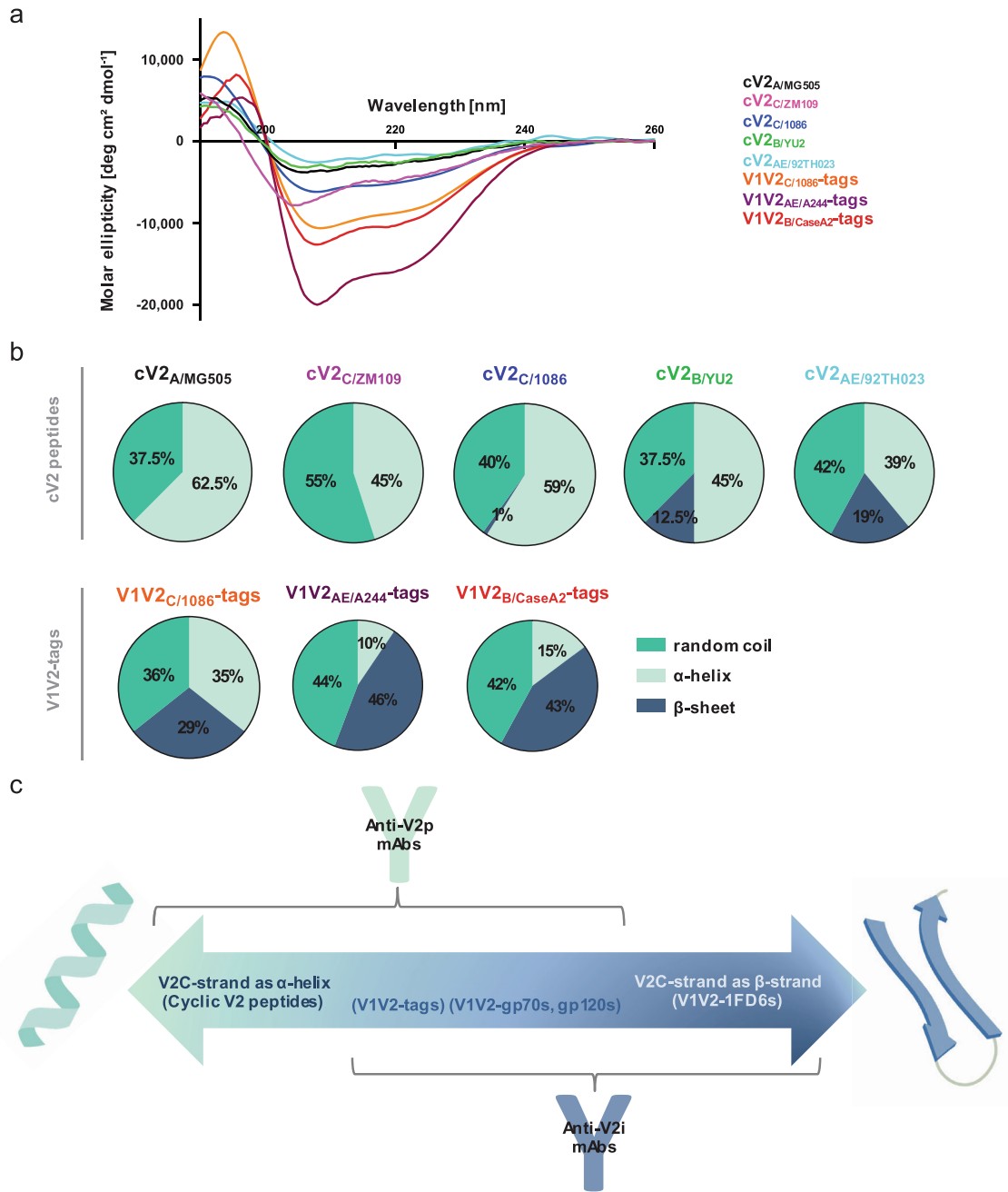

**Fig. 2 Structural conformations of the V1V2 domain. a** Circular dichroism spectra of five cyclic V2 peptides and three V1V2-tags proteins. **b** Pie charts depicting the percentage of secondary structure (random coil [turquoise], α-helix [light green], or β-sheet [dark blue]) determined by the circular dichroism spectra and calculated by CD-FIT software (https://www.ruppweb.org/Xray/comp/cdfit.htm). **c** Schematic diagram created with BioRender.com showing the alterative structures of the V2 C-strand (V2C) between an α-helix (left) and a β-strand (right). According to the proportion of each molecule present in the two conformations, the antigens will react preferentially with V2p Abs (left), V2i Abs (right), or both (middle). Abbreviations as in Fig. 1.

fused with an N-terminal Ig leader sequence and C-terminal avi- and his-tags (V1V2-tags), the α-helix/random coil and β-strand proportions of V1V2 or V2 Ags studied ranged from 54–71% and 29–46%, respectively. These findings are consistent with those of Rao et al.[61] and Wibmer et al.[16], where additional cV2 peptides exhibited a propensity to form random coil/alpha-helices.

Given the preferential specificity of V2p mAbs for cV2 peptides and of V2i mAbs for V1V2-1FD6 scaffold proteins, reactivity with cV2 peptides was used as a hallmark for the presence of polyclonal V2p Abs in plasma from infected and immunized animals, while V1V2-1FD6 molecules were used as a marker to

identify V2i plasma Abs. Gp120, V1V2-tags and V1V2-gp70 reagents reacted with both V2p and V2i mAbs (Fig. 1a) and therefore were not used to identify V2p and V2i Abs. This paradigm is represented diagrammatically in Fig. 2c showing the alternative α-helix and β-strand configurations of the V2C region, the mAbs that preferentially bind to each, and the Ags that exist in either or both configurations.

**Patterns of plasma Ab binding to V2 and other Env epitopes differ between infected and immunized rhesus macaques.** We

**Table 1 HIV-1 vaccine regimens studied in *M. mulatta*.**

| Immunization designation | Prime (*env/gag* genes) | Protein boost | Adjuvant | Schedule | Refs. |
|---|---|---|---|---|---|
| DNA + gp120 | DNA (AE/92TH023 gp160 and SIV gag) | gp120 (B/MN + AE/A244) | Adjuplex | Wk 0: DNA + gp120s<br>Wk 4: DNA + gp120s<br>Wk 12: DNA + gp120s<br>Wk 20: DNA + gp120s | 62 |
| SAd7 + gp140 | Simian Adenovirus 7 (SAd7 includes transgenes for C/1086 gp150 HIV-1 *env* and SIVmac239 *gag*) | gp140$_{C/1086}$ trimer + GBV-C E2[a] | Adjuplex | Wk 0: SAd7<br>Wk 4: SAd7 + protein<br>Wk 16: Protein | 53 |
| DNA + V1V2-scaffold | DNA (C/ZM53 gp120) | V1V2$_{C/ZM53}$-2F5K<br>V1V2$_{C/ZM109}$-TTB<br>V1V2$_{AE/A244}$-2J9C | Adjuplex | Wk 0: DNA + V1V2-scaffold cocktail<br>Wk 8: DNA + V1V2-scaffold cocktail<br>Wk 20: DNA + V1V2-scaffold cocktail | 22 |

[a]Human pegivirus GBV-C E2 glycoprotein.

performed experiments to determine the relative levels of these Abs induced by infection of NHPs with either clade C Tier 1 SHI-V$_{1157ipEL-p}$ or clade C Tier 2 SHIV$_{1157-ipd3N4}$ and by vaccination of NHPs with one of three distinct immunization regimens. All immunizations consisted of co-immunization or prime/boost strategies in *M. mulatta* with the same adjuvant (Adjuplex) as shown in Table 1: the "DNA + gp120" group is comprised of animals in which DNA encoding for gp160$_{AE/92TH023}$ and SIV$_{mac239}$ gag was delivered concomitantly with HIV Env gp120 proteins (clades B and E) on weeks 0, 4, 12, and 20[62]; the "SAd7 + gp140" group received simian adenovirus 7 (SAd7) carrying genes for HIV clade C Env gp150$_{C/1086}$ on weeks 0 and 4, and gp140$_{C/1086}$ and GBV-C E2 proteins on weeks 4 and 16[53]; and the "DNA + V1V2-scaffold" group was immunized with HIV clade C *env* gp120$_{C/ZM53}$ DNA at weeks 0, 8, and 20 along with three V1V2-scaffold recombinant proteins, V1V2$_{C/ZM53}$—2F5K, V1V2$_{C/ZM109}$-TTB and V1V2$_{AE/A244}$—2J9C[22,62,63].

To identify the patterns of V2 Ab specificities induced by infection, specimens were obtained at necropsy from the animals infected with Tier 1 SHIV$_{1157ipEL-p}$ and Tier 2 SHIV$_{1157-ipd3N4}$ at weeks 18 and 11 post last challenge, respectively. For immunized animals, blood was drawn 2 weeks after the last immunization. Plasma specimens were initially screened for binding to the Ag panel described in Fig. 1a and Supplementary Fig. S2 using a plasma dilution of 1:200 (Fig. 3 and Supplementary Fig. S3). These data are complemented by titration of plasma using dilutions of 1:100–1:150,000 for SHIV-infected animals (Supplementary Fig. S4a–b) and for "DNA + gp120" immunized NHPs (Supplementary Fig. S4c).

Plasma from the SHIV-infected animals and animals in all the immunized groups reacted with gp120 glycoproteins from clades A, B, and C; they also reacted strongly with V3$_{consC}$ even in the animals that had not been exposed to clade C immunogens, i.e., the "DNA + gp120" group. Antibodies specific for C5, which are among the highest titered Abs in infected humans[5,64], were elicited in infected NHPs. Notably, among the immunized NHPs, only those in the "DNA + gp120" group mounted a C5 Ab response (Fig. 3 and Supplementary Fig. S4c). These data suggest that C5 Abs are poorly induced by the gp120$_{C/ZM53}$ DNA (DNA + V1V2-scaffold group) and by the gp140 protein boost (SAd7 + gp140 group). Given the conservation of C5, it is probable that if these animals, immunized with a clade C gp140, could have mounted a C5 Ab response, they would have reacted with the C5$_{consC}$ peptide. A dampened exposure of C5 in gp140 proteins has been attributed to conformational changes in gp120 due to the lack of membrane and cytosolic gp41[65].

Further examination of the data indicates that strong responders had high levels against all Ags tested, and conversely, for weak responders (Supplementary Fig. S3).

There is a remarkable paucity of V1V2-specific Ab responses in the Tier 2- and Tier 1-SHIV-infected animals 11 and 18 weeks after the last challenge dose, respectively (Fig. 3); this stands in contrast to the uniform Ab responses to gp120 from clades A, B, and C and to a V3 peptide heterologous to the infecting strains. These data suggest that if there was a substantial Ab response to V2 epitopes in the infected NHPs, it would have been detected given the similarities at key residues between the V2p regions found in the two clade C V2 peptides studied here and the similar region in the SHIV strains used to infect the NHPs (Supplementary Table S1). In addition, it is unlikely that mutations that might have occurred in the V2 region after 18 weeks of SHIV infection would account for a loss or gain of cross-reactive Abs.

All immunized animals displayed Abs to Ags carrying V2 or V1V2. The most extensive V1V2 responses were obtained in the "DNA + V1V2-scaffold" immunized group which displayed Abs reactive with all 19 of the V2 and V1V2 Ags tested. The next most V2-responsive group was the "SAd7 + gp140" group (reactive with 12/19 V2 and V1V2 Ags), followed by the "DNA + gp120" group (reactive with 11/19 Ags).

Interestingly, V2p Ab responses in all three immunized NHP groups were highly reactive with the linear peptides spanning V2(AA166-185)$_{AE/A244}$ and V2(AA172-191)$_{AE/A244}$ which corresponds to the region covering the V2C-strand that includes the residues at positions 169 and 181 that were the key signatures of escape from vaccine efficacy in the RV144 clinical trial. The V2p Abs were cross-reactive with peptides from heterologous strains and clades. For example, plasma from the "SAd7 + gp140" group which received HIV immunogens from clade C strain 1086 reacted with V2 peptides from the homologous and heterologous strains C/1086 and C/ZM109, and with V2 peptides from the heterologous clade AE strain 92TH023 (Fig. 3 and Supplementary Fig. S5). Though only a single V2 peptide from clades A and B were tested, there was poor or no reactivity with these peptides in either infected or immunized animals, including those in the "DNA + gp120" group which received the gp120$_{B/MN}$ protein. In contrast to the reactivity with V2C-strand peptides, there was little or no Ab reactivity in the plasma of infected or immunized NHPs with a peptide that corresponds to the V2D-strand (V2(AA190-209)$_{AE/A244}$). These data demonstrate that the vaccine-induced Abs from all three vaccine regimens tested focused the Ab response on the V2C strand in its α-helix/random coiled configuration.

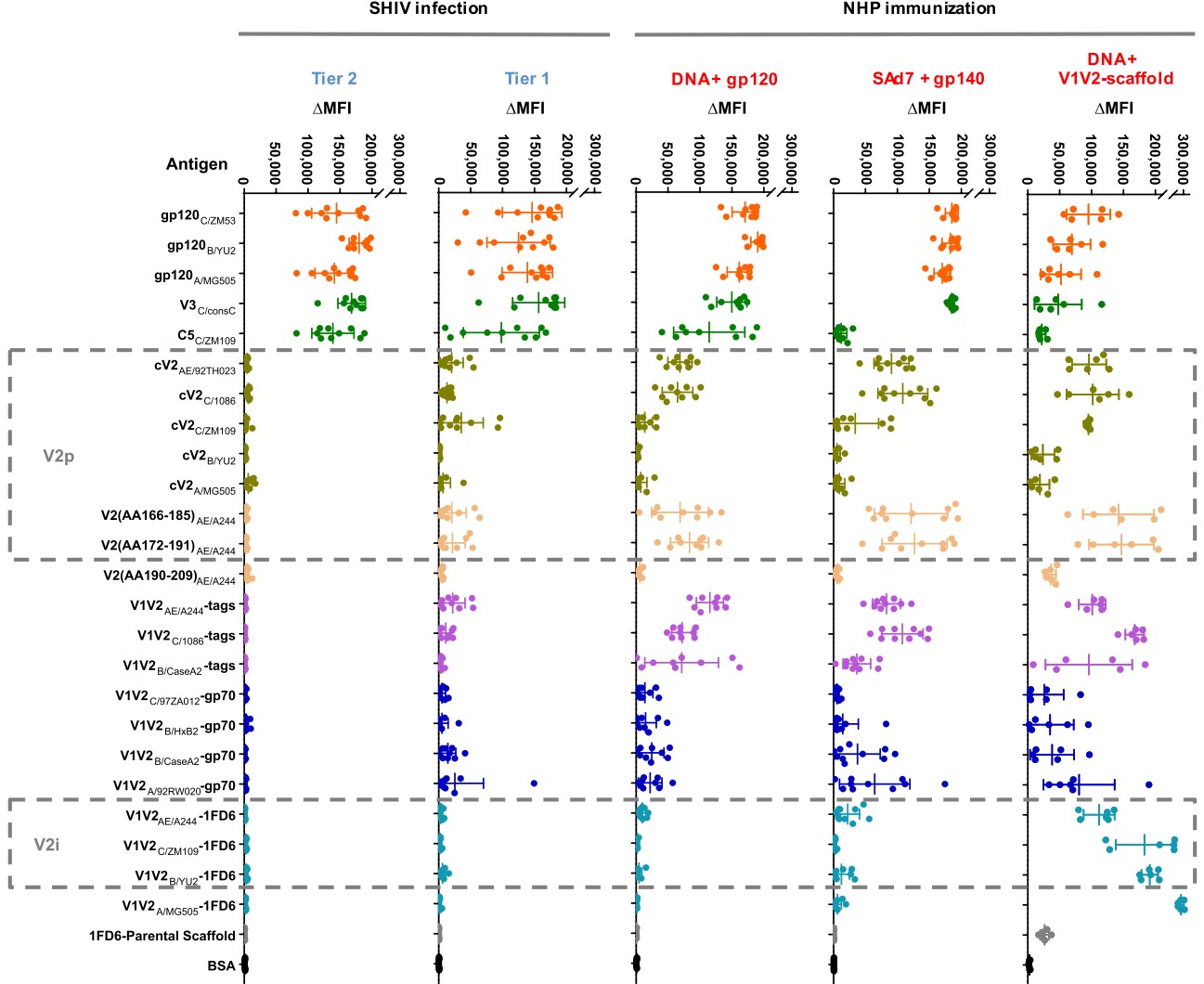

**Fig. 3 Dot plot of Ab binding activities in plasma from infected and immunized NHPs assessed in the multiplex bead Ab binding assay.** Columns from left to right: Tier 2 SHIV$_{C/1157-ipd3N4}$-infected NHPs ($n = 10$); Tier 1 SHIV$_{C/1157ipEL-p}$-infected NHPs ($n = 10$); NHPs immunized with "DNA + gp120" ($n = 9$); NHPs immunized with "SAd7 + gp140" ($n = 10$); NHPs immunized with "DNA + V1V2-scaffolds" ($n = 6$). Each group was composed of biologically independent plasma specimens. Binding activity of plasma was measured against the same set of 24 Env antigens used in Fig. 1a; each row lists a tested antigen, and the rows are grouped by antigen type (peptide; gp120, etc.). Boxes with dashed lines denote antigens reactive with V2p- or V2i-specific mAbs. Beads identified as "1FD6 parental scaffold" (devoid of the V1V2 insert) and BSA (bovine serum albumin) were used as negative controls. Each dot represents the normalized mean fluorescence intensity (ΔMFI) generated with a plasma specimen diluted 1:200 from a single animal. Error bars indicate mean and standard deviation (SD). Colors of the dots represents a type of antigen (orange = gp120s, dark green = non-V1V2 linear peptides (V3 and C5), olive green = cV2 peptides, cream = linear V2 peptides, purple = V1V2-tags, dark blue = V1V2-gp70s, light blue = V1V2-1FD6s, gray/black = negative controls). Intensity of the reactivity is shown as the mean MFI calculated from experiments normalized on the basis of the reactivity of the mAb pool in each experiment from which background (PBS-TB) was subtracted. Data are represented as mean with error bars that indicate standard deviation (SD). Experiments were performed at least twice and in each experiment, samples were tested in duplicate. Source data are provided as a Source Data file. Abbreviations as in Fig. 1.

As noted, the primary variable in RV144 associated with the reduced risk of infection was Abs reactive with V1V2-gp70$_{B/CaseA2}$ (refs. [2–4]), a reagent recognized by both V2i mAbs and V2p mAbs (Fig. 1a and Supplementary Figure S2). A V2i-specific mAb was also associated with the partial control of SHIV$_{Bal}$ infection[66]. Therefore, it is noteworthy that among the five groups of animals studied, only the "DNA + V1V2-scaffold" group mounted a V2i Ab response as shown by the reactivity of the plasma from animals in this group with V1V2-1FD6 constructs.

**V1V2-scaffold-induced V2p and V2i-specific antibodies mediate Fc-dependent functions**. To dissect the functions of vaccine-induced Abs that are specific for V2p and/or V2i, two assays were

used: (a) binding of the C1q component of complement to the Fc fragment of Abs bound to cV2 peptides (cV2$_{C/1086}$ and cV2$_{AE/92TH023}$) or to V1V2-1FD6 scaffold proteins (V1V2$_{C/ZM109}$−1FD6 and V1V2$_{B/YU2}$−1FD6) which denotes the initiation of complement activation, and (b) ADCP initiated by beads coated with the same Ags. Both of these functions are associated with non-neutralizing control of HIV and SIV infection[67,68]. Results from these assays are represented as AUC values (Fig. 4) derived from titration curves (Supplementary Fig. S6).

C1q binding by V2p and V2i Abs. Despite the fact that V2p-specific binding Abs were detected in the plasma of all three groups of immunized animals (Figs. 3 and 4a), V2p Abs recognizing cV2 peptides bound C1q poorly, if at all (Fig. 4b

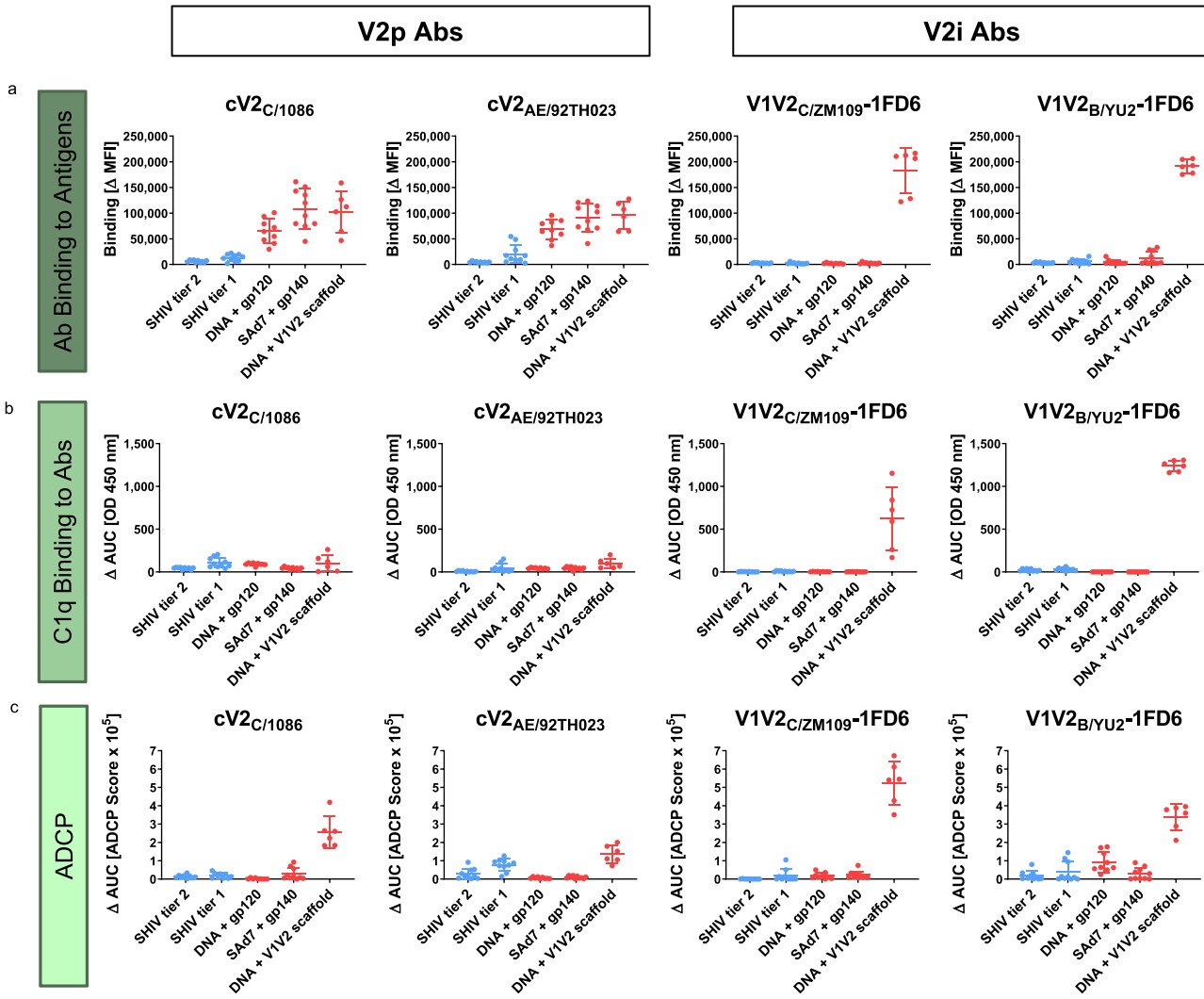

**Fig. 4 Antibody binding, C1q binding, and ADCP responses of NHPs infected with SHIV (blue) ($n = 20$) or immunized with one of the three immunization regimens (red) ($n = 25$).** Each data point is derived from a biologically independent plasma specimen. Each panel shows results from a different assay: **a** multiplex bead Ab binding, **b** C1q binding, and **c** ADCP. Activities were measured using cV2 peptides (reactive with V2p Abs) or V1V2-1FD6 scaffold proteins (reactive with V2i Abs). Binding values in panel **a** are ΔMFI as shown in Fig. 3. Values in panels **b** and **c** represent area-under-the-curve (ΔAUC) and were calculated for C1q from two-fold plasma titrations ranging from 1:50–400, and for ADCP from five-fold dilutions ranging from 1:50–781,250 (Supplementary Fig. S6). ΔAUC values were calculated as the AUC from each NHP group 2 weeks after immunization or 11 (SHIV Tier 2) or 18 (SHIV Tier 1) weeks after infection from which the pre-bleed background response was subtracted. The mean of all animals within each group and the standard deviation are indicated. C1q titrations were performed once and ADCP experiments were performed at least twice. Source data are provided as a Source Data file.

and Supplementary Fig. S6a). These data indicate either that the V2p Abs are of an isotype that does not activate complement or, more likely, that binding to peptides does not permit the aggregation of Abs needed to initiate the complement cascade.

In contrast, when C1q binding to Abs was measured using V1V2-1FD6 Ags, the animals receiving the "DNA + V1V2-scaffold" regimen were the only immunized animals with V2i-specific Abs and the only animals able to bind C1q. The C1q binding was observed with V1V2-1FD6 Ags derived from the homologous C/ZM109 immunogen used in the "DNA + V1V2-scaffold" group, and from a heterologous clade, B/YU2 (Fig. 4b and Supplementary Fig. S6a).

Plasma of both Tier 1 and Tier 2-infected animals had strong responses to gp120 but low levels of V2p Abs and no detectable V2i Abs (Fig. 3). C1q binding by gp120-specific Abs from Tier 2-infected animals was low but detectable whereas no plasma V2p or V2i Abs from this group were able to bind to C1q. Plasma

from Tier 1-infected animals showed only weak binding of C1q to V2p Abs, but no C1q binding to either gp120 or V2i-specific Abs (Supplementary Fig. S6a). In summary, of the Abs tested in the plasma of the two groups of SHIV-infected NHPs and the three groups of immunized animals, primarily those receiving the "DNA + V1V2-scaffold" vaccine regimen responded with V2 Abs able to activate the complement cascade.

**ADCP mediated by V2 Abs.** As previously reported, all three HIV-1 Env vaccine regimens were able to induce Abs with Fc-dependent effector functions including ADCC against SHIV$_{SF162p3}$[22,62,63], ADCP against gp120$_{B/BaL}$[62], V1V2$_{C/ZM09}$-1FD6, V1V2$_{C/ZM109}$-TTB and V1V2$_{C/ZM53}$-2F5K[22,63], and ADCVI against SHIV-1157ipEL-p[53].

Detection of ADCP activity was observed in the plasma of animals immunized with the "DNA + V1V2-scaffold" regimen and was seen with beads coated with V1V2-1FD6 proteins and with beads coated with cV2 peptides (Fig. 4c and Supplemental

Fig. S6b). Neither infected NHPs nor animals vaccinated with the other two immunization regimens displayed ADCP activity. Thus, only the V2-targeting immunization regimen induced Abs that could mediate both C1q binding and ADCP activities since only immunization with the V2-targeting regimen (DNA + V1V2-scaffolds) induced V2p and V2i Abs, only this group of immunized animals displayed ADCP activity, and the ADCP activity correlated strongly with the presence of V2i Abs (Fig. 4 and see below).

**Full parameter analysis reveals Ab signatures that are distinct between NHPs infected with SHIVs or immunized with different regimens.** The compiled data from all 45 NHP plasma samples, generated by the multiplex Ab binding, C1q and ADCP assays are summarized in the cluster heatmap shown in Fig. 5. The data for each animal are shown in a separate row. The data from Ab binding, C1q binding, and ADCP assays using different Ags are depicted in each column.

Hierarchical clustering of the animals reveals that the NHPs in the three immunization groups (pink) cluster separately from the infected animals (blue, see dendrogram on left of heatmap) and that there is an intermixing of Tier 1 and Tier 2 SHIV-infected animals as well as an intermixing of the animals receiving the "DNA + gp120" and "SAd7 + gp140" vaccine regimens. The animals in the "DNA + V1V2-scaffold" group are clearly distinct (Fig. 5a, b). Additional principal component analyses show the separate clustering of each of the five groups of animals, with the "DNA + V1V2-scaffold" group being the most unique (Fig. 5c and Supplementary Fig. S7).

Cluster analysis of the binding data revealed by the dendrogram at the top left of Fig. 5a shows four families of Ags composed of (i) gp120, V3, and C5; (ii) primarily V2 peptides; (iii) primarily V1V2-1FD6; and (iv) primarily V1V2-gp70. This representation of the binding data indicates that (a) the "DNA + V1V2-scaffold" group shows *decreased* binding to gp120, V3, and C5 (family i), (b) the immunized groups but not the infected groups have mounted a strong response to V2p Ags (family ii), (c) only the "DNA + V1V2-scaffold" group makes Abs that recognize the V2i Ags (family iii), and (d) all groups make Abs that recognize Ags that co-exist in the V2p and V2i configurations, although this reactivity is weakest in infected animals (family iv).

**V1V2 immune profiles cluster separately from immunodominant responses to gp120, C5, and V3 epitopes.** To examine the interrelationships between Ab binding and biological activity, multi-parameter linear correlation analyses were performed (Fig. 6) with statistical analyses detailed in Supplementary Figure S8). Three clusters were identified showing strong positive correlations (outlined with boxes in yellow = cluster 1, red = cluster 2, and orange = cluster 3, Fig. 6), and an additional two clusters show negative correlations (outlined by boxes in black = cluster 4 and blue = cluster 5 in Fig. 6). Cluster 1 shows strong positive correlations between binding of Abs to gp120 glycoproteins and the consensus C V3 peptide. Cluster 2 shows strong positive correlations between Abs binding to V1V2-gp70 and V2 peptides, i.e., Ags that bind primarily to V2p Abs. Cluster 3 shows strong positive correlations primarily between Abs binding to V1V2-1FD6 (which bind V2i Abs) and Abs that mediate ADCP and complement activation. This correlation analysis supports the data shown above (Figs. 4 and 5a) indicating that the only group of animals whose plasma Abs displayed C1q binding and ADCP was the "DNA + V1V2-scaffold" group in which the vaccine response resulted in both V2p and V2i Abs.

Clusters of negative correlations were also identified: cluster 4 shows strong negative correlations between Abs that display V1V2-1FD6-binding, C1q-binding, and ADCP activity vs. Abs that bind to gp120 and V3. Cluster 5 shows negative correlations of anti-C5 Abs vs. Abs that mediate C1q, ADCP, and that bind to V1V2 Ags. This correlation analysis supports the data shown in Figs. 4 and 5a in which, again, it was the "DNA + V1V2-scaffold" group which showed the strongest V1V2 responses, C1q binding and ADCP, but the poorest Ab levels to gp120, V3, and C5.

A correlation network analysis of data from the multiplex Ab bead binding assay (Fig. 7) highlights the anti-V2 Ab responses which are strongly positively correlated with each other but are negatively correlated with Ab responses to gp120, V3, and C5.

## Discussion

Identifying virus epitopes that induce protective Abs is the key to designing effective vaccines. The rapid identification of the receptor-binding domain of the SARS-CoV-2 Spike protein and the production of highly effective vaccines based on multiple platforms in 2020 has been remarkable. In contrast, the HIV field has labored for over 40 years to accomplish a similar feat and remains confronted with the hurdles imposed by a trimeric envelope protein on which the sites of vulnerability are occluded by glycans, along with extensive mutational variation and extreme conformational flexibility. This presents a conundrum in terms of which epitope(s) should be and can be targeted to elicit a protective immune response to HIV.

Several studies in the literature suggest that the V1V2 domain of Env contains epitopes worth targeting with a vaccine. These include quaternary epitopes targeted by such Abs as PG9 and PGT145 (reviewed in Kwong and Mascola[69]), as well as epitopes that are not dependent on the trimeric structure of Env, such as those targeted by V2i mAbs such as 2158 and 697, and by V2p mAbs such as CH58 and CAP228-19F[15,19,20,70]. Data support the hypothesis that Abs directed at the V1V2 domain of gp120 contributed to a reduced risk of HIV infection in humans[2–5,71], and similarly have been implicated in the reduction and control of infection of NHPs with SIV[70,72–78]. Additional studies in macaques infected with various strains of SHIV also appear to support this hypothesis but are not definitive[8,53,79]. Using these studies as a foundation, and samples from vaccinated NHPs and non-vaccinated SHIV-infected NHPs, we developed a molecular toolkit of curated antigens representing multiple epitopes within V1V2, gp120 proteins from various clades, and V3 and C5 peptides to characterize and parse immune responses in NHPs. In particular, we sought to identify Ab responses induced by three different immunization regimens and by infection with Tier 1 and Tier 2 SHIVs in order to determine the extent to which these responses targeted the V1V2 domain, and to compare the patterns of V1V2 Abs for quantity, function, and fine specificity.

While broadly neutralizing V2q and V2qt Abs appear in <2% of HIV-infected individuals[80], they have yet to be induced by any vaccines tested in humans or animals. In contrast, V2p and V2i Abs are present in the serum of 30–84% of infected individuals[81–84] and have been induced with vaccines in humans[1,85,86]. As we have shown here and previously[22], Abs of these specificities can be induced with vaccines in NHPs, and they are cross-clade reactive, and exhibit anti-viral activities, although, at best, are poorly neutralizing. The current study describes methods for measuring the presence and levels of Abs specific for V2p and V2i epitopes in plasma and for assessing the anti-viral activities of these two types of Abs. Here, we have also evaluated an effective approach that selectively induces V2p and V2i Abs.

Heretofore, reagents have not been available to easily distinguish between the various types of V2 Abs in plasma. For this, we

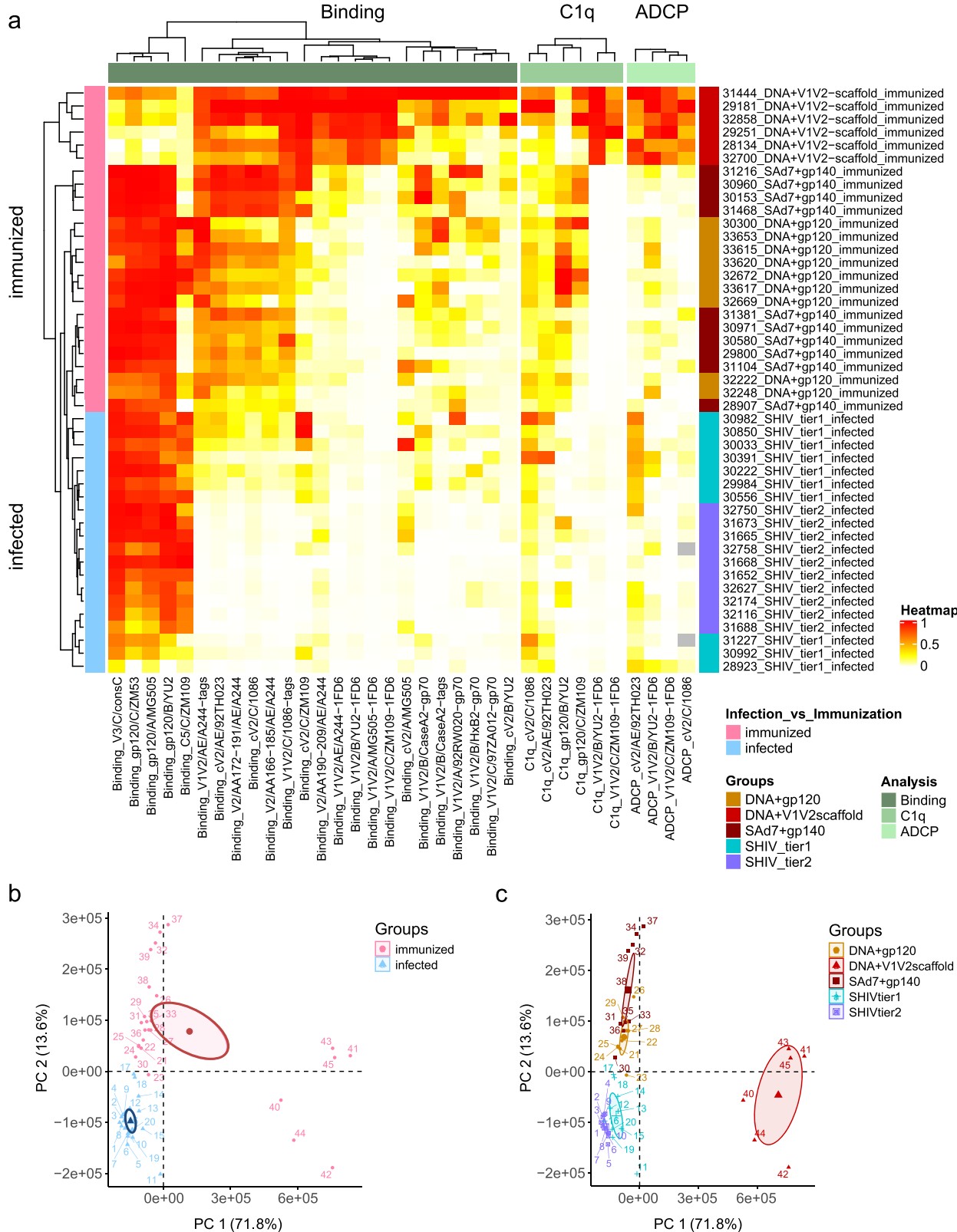

**Fig. 5 Analysis of data from all 45 animals and all three types of assays. a** Heatmap summarizing the humoral immune responses in immunized versus SHIV-infected NHPs. Immune responses shown in Figs. 3 and 4 and Supplementary Figs. S3–S6 were normalized and color-coded as indicated in the heatmap legend. Columns represent immune responses grouped according to Ab binding, C1q binding, and ADCP and are clustered within each parameter (Ab binding, C1q binding, or ADCP) measured using the antigens shown at the bottom of the heatmap. Rows represent single animals, each identified by number, vaccine regimen, or SHIV-strain used for infection. The heatmap is shown with hierarchical clustering according to the three dendrograms at the top and the one to the left. **b**, **c** Principal component analyses were performed using all values from panel **a** which are colored in panel **b** as immunized (pink) versus infected (blue), or in panel **c**, according to NHP groups as indicated in the legend. The axes show the Principal Component 1 and 2 (PC 1 and PC 2). The ellipses indicate 95% confidence interval of the means.

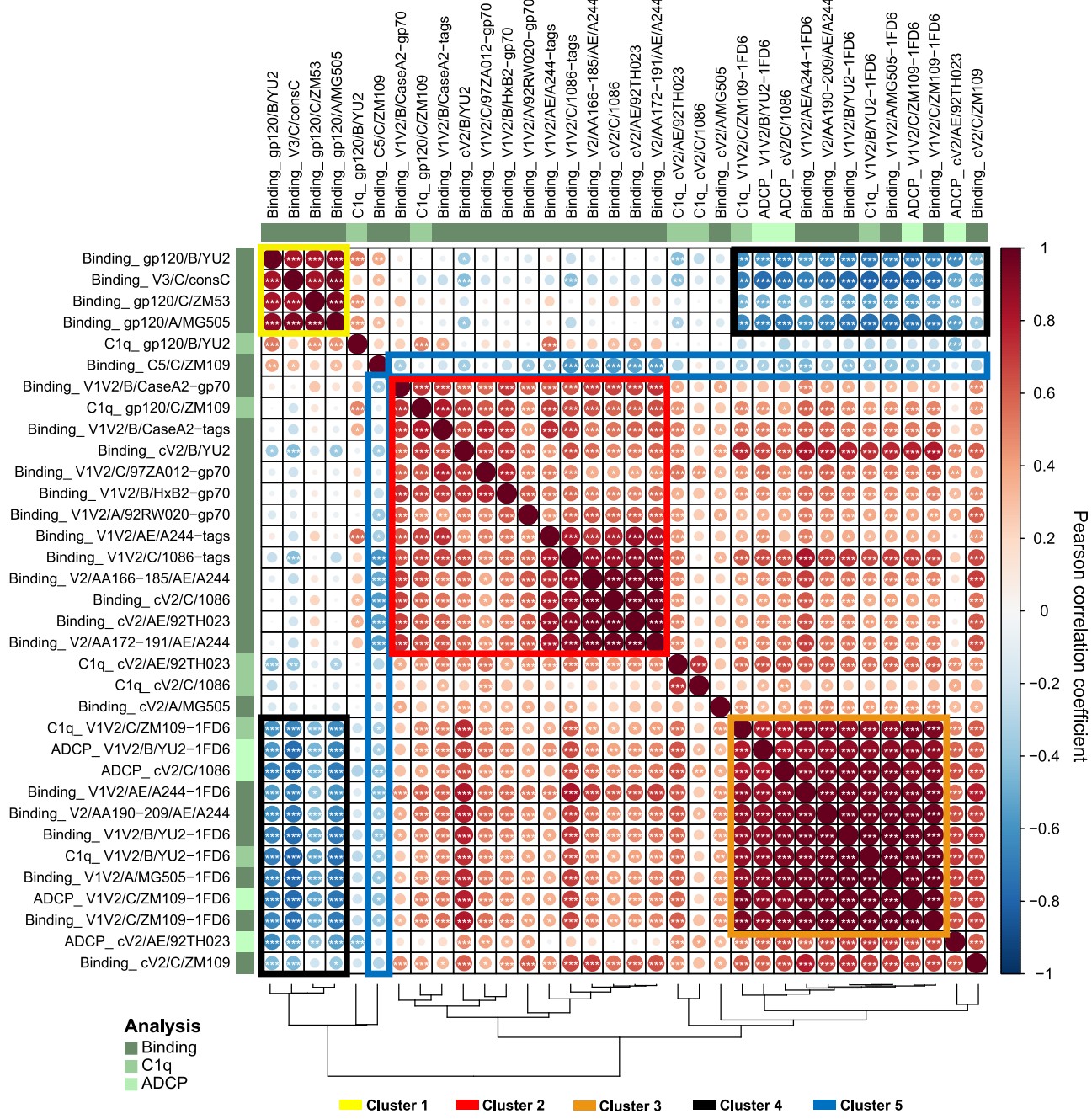

**Fig. 6 Correlation analysis of Ab binding, C1q binding, and ADCP against various HIV-1 Env antigens.** Correlogram summarizing pairwise correlations among indicated parameters for all 45 studied NHPs. In the correlogram, the magnitude of the correlation coefficient $r$ is color-coded as per the spectrum shown on the right with red colors representing positive and blue colors negative correlations between any two parameters. Asterisks within each cell indicate statistically significant correlations (*$p < 0.05$, **$p < 0.001$, ***$p < 0.005$; two-tailed $p$ values from Pearson correlation). The correlogram is shown with hierarchical clustering according to the dendrogram at the bottom. The five most prominent correlation clusters are highlighted with boxes and numbered as indicated in the legend. Source data are provided as a Source Data file.

developed a panel of Ags bearing the V2 sequences for the V1V2 domains from several clades and strains and characterized them in terms of their ability to cross-react with 11 mAbs that target the V2p, V2i, V2q, and V2qt epitopes (Fig. 1a and Supplementary S2 and Supplementary Table SI). The reactivity of these Ags in a multiplex Ab binding assay indicated that V2p mAbs bound to V2 peptides which present the V2C-strand as an α-helix/coil,[15,23,87] while V2i mAbs bind preferentially to V1V2-1FD6 Ags in which V2C is most frequently present as a β-strand as part of the V1V2 β-barrel[13,14]. As we show, there are exceptions: reagents based on clade AE present an unusual pattern: the V1V2-1FD6 scaffold protein and the V2 peptides derived from clade AE can assume both the α-helix/coil and β-strand conformations as shown by their ability to be bound by both V2p and V2i mAbs (Fig. 1b). This may relate to the unusual ability of the clade AE gp120$_{A244}$ to induce in humans particularly strong V2-specific responses directed to both V2p and V2i[2,88], and the ability of the "DNA + V1V2-scaffold" group of immunized NHPs (which were boosted with three V1V2-scaffolds, including V1V2$_{AE/A244}$-2J9C) to induce both V2p and V2i Abs (Fig. 3).

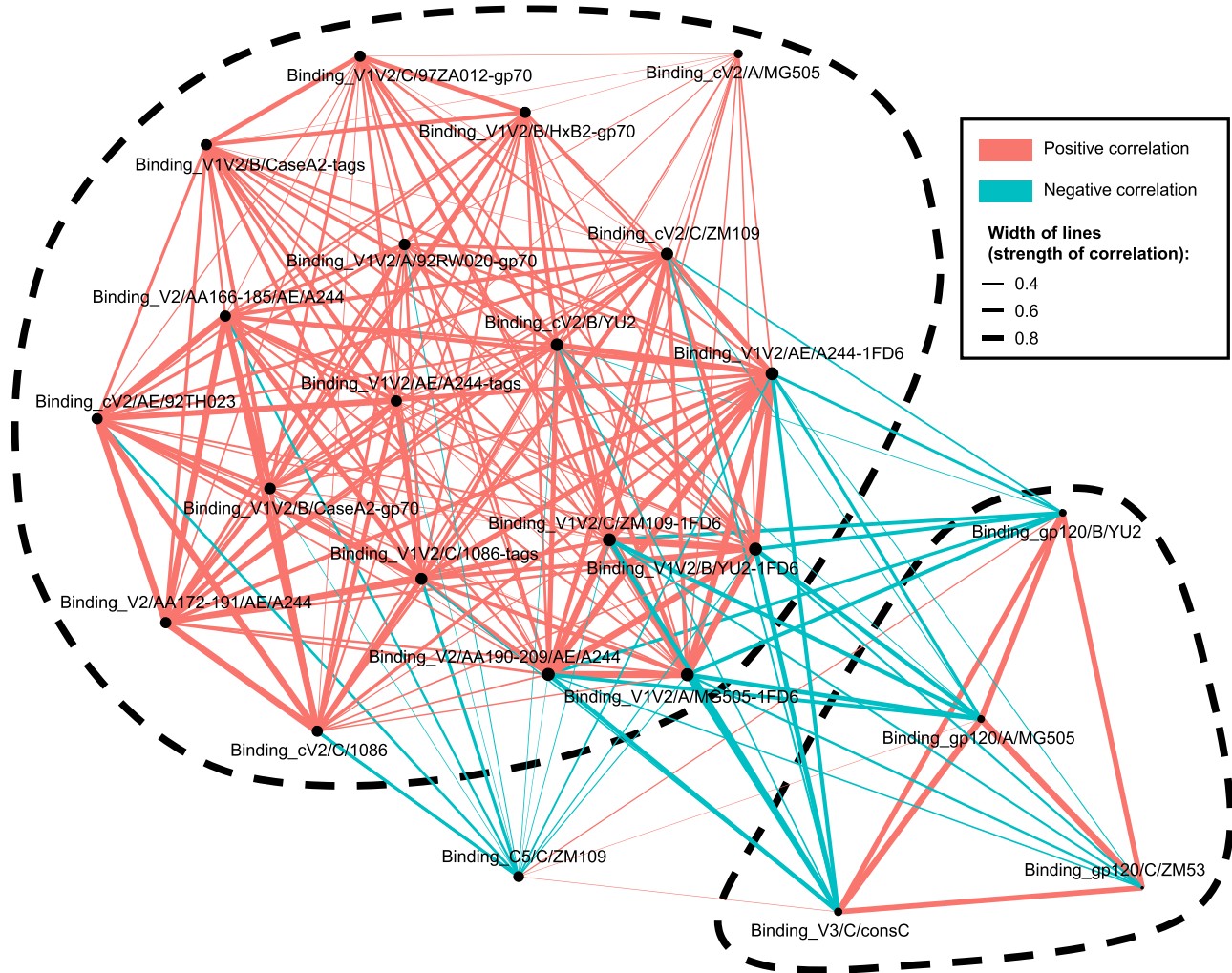

**Fig. 7 Correlation network analysis of Ab binding responses among all studied NHPs.** Nodes represent Ab responses against each specific designated antigen, and red and blue lines represent positive and negative correlations between connected parameters, respectively. Only significant correlations ($p < 0.05$; two-tailed $p$ values from Pearson correlation, here not adjusted for multiple comparison; multiplicity correction is shown in Supplementary Fig. S7) are displayed. Line thickness corresponds to the strength of the correlation coefficient ($r^2$). Node size, indicated by the size of the black closed circles at each node, corresponds to the sum of correlations of the respective Ab response. The two most prominent correlation clusters are encircled with dashed lines.

In RV144, the inverse CoR associated with the level of V2 Abs was identified using V1V2$_{B/CaseA2}$-gp70[2–4]. Since V1V2-gp70 reagents react with both V2p and V2i Abs (Fig. 1a), the primary RV144 CoR with V1V2$_{B/CaseA2}$-gp70 did not indicate which Ab type, V2p and/or V2i, was correlated with reduced risk, although the subsequently identified CoR with V2 peptides supports the hypothesis that V2p Abs were involved[5]. Since passive immunization of NHPs with V2i mAb 830A followed by challenge with SHIV$_{BaL}$ led to reduced virus levels in plasma and peripheral blood mononuclear cells and decreased viral DNA in lymphoid tissues, there is evidence that V2i Abs can contribute to control of infection[66]. Additionally, though most studies of V2 correlates of protection in NHPs against SIV and SHIV did not test for V2i Abs, Abs specific for V2 peptides were correlated with protection and/or control (reviewed in Zolla-Pazner[8]). Two additional active immunization studies support a protective role for V2 Abs: higher V1V2-specific binding antibody titers correlated with viral control in the SAd7 + gp140 vaccine group[53], and in a challenge study of the nine animals in the "DNA + V1V2" group, four of nine NHPs ($p = 0.0497$) immunized with a V2-focusing regimen showed an absence of plasma vRNA, cell-associated vDNA in

PBMCs, and lymphoid-associated virus over a 9-week period following the final challenge dose with the heterologous Tier 1 SHIV$_{BaL.P4}$[62]. In the latter study, the predominant V2 Ab response was directed at V2 peptides, with little or no response to V1V2-1FD6, demonstrating a predominant effect of V2p Abs. However, the V2 responses did not differ in the tight controllers compared with the non-controllers, leaving open the question of the role of the V2p-directed Abs using this regimen.

The Luminex data presented above show that all three immunization regimens examined here induced V2p Abs but only the "DNA + V1V2-scaffold" immunization regimen gave robust Ab responses to both V2p and V2i Ags (Fig. 3). Importantly, all three immunization regimens induced cross-clade reactive Ab responses, with reactivity to Ags carrying V1V2 and V2 segments from clades A, B, C, and AE. The broad cross-reactive binding of the vaccine-elicited V2 Abs underscores shared antigenic features[89] and the common structure that previous studies have shown the V1V2 domain to assume[13,14]. Notably, boosting with the V1V2-scaffold proteins elicited only weak responses to gp120, V3, and C5 (Figs. 3 and 5). This is an indication that the use of the V1V2-scaffold proteins focused the

immune response on the V2 region, which was the intent of the design of this immunization regimen[22,37,55].

It is noteworthy that, with the sole exception of Abs to C5, immunization by the three vaccine regimens induced responses that were equal to or stronger than those induced by infection (Figs. 3 and 5). This indicates that vaccines can induce immune responses that are qualitatively and quantitatively different compared to infection, and this is especially true for V2 Ab responses. Since infection does not lead to subsequent protection, as shown by the ability of HIV to superinfect already infected individuals (reviewed in Redd et al.[90]), effective HIV vaccines will have to do better than Nature. The Ab responses elicited by HIV vaccines will need to preferentially target epitopes that elicit functional, cross-reactive, and durable Ab responses. The fact that native Env trimers presented on virions in the course of infection are not able to induce these features suggests that epitope-targeting immunogens, such as the ones tested in the "DNA + V1V2-scaffold" group, may be particularly useful constructs. It remains to be investigated whether responses induced by these regimens are protective.

In the data presented, only the V2-directed immunization regimen induced Abs with the Fc-mediated effector functions of ADCP and complement activation. Assessing the combined datasets of Ab binding and functionality, an inverse immune pattern was observed for V2 responses compared to those for gp120, V3, and C5 (Figs. 5–7). These data suggest that Abs to immunodominant epitopes including V3 and C5 that are induced by whole Env vaccines may divert immune responses from less immunogenic epitopes such as those responsible for the induction of functional and potentially protective Ab responses, including those specific for V2p and V2i.

We and others have shown that both V2i and V2p mAbs can mediate ADCP and ADCC[20,22,37,91], and that both ADCP and ADCC have been correlated with reduced risk of HIV and SIV and SHIV infection in NHPs[8]. Indeed, both ADCC and ADCP were analyzed in the monkeys receiving the "DNA + V1V2-scaffold" vaccine and the responses were robust and long-lasting[22,63].

Taken together, the findings of this study suggest that the patterns of induced V1V2-specific Abs differ significantly between SHIV infection and immunization in NHPs, that immunization can induce more potent binding responses to this region of Env than infection, and that effective induction of functional V2p and V2i Abs benefits from targeting the V1V2 domain with V1V2-scaffold proteins. Moreover, since V1V2 Abs display a variety of Fc-mediated anti-viral functions that correlate with protection from HIV, SIV, and SHIV, data showing that V2p and V2i Abs can easily be induced is of particular significance. Given that these data indicate that the levels of V1V2-specific Abs can be enhanced by epitope-targeting vaccines, they constitute an important consideration in terms of moving the field toward the development of an efficacious HIV vaccine. Finally, the type of comprehensive analysis of Ab fine specificity targeting Env epitopes will be important for discriminating between immunization protocols and perhaps for establishing go/no-go criteria for clinical trials.

## Methods

**Study approval.** Macaque studies were performed at the Oregon National Primate Research Center (ONPRC) in Beaverton, OR, USA. The ONPRC is accredited by the American Association for the Accreditation of Laboratory Animal Care International, and adheres to the Guide for the Care and Use of Laboratory Animals and the U.S. Public Health Service Policy on the Humane Care and Use of Laboratory Animals. The Oregon Health & Science University (OSHU) West Campus Institutional Animal Care and Use Committee (IACUC) approved all macaque studies.

**Animals.** A total of 45 adult male ($n = 32$) and female ($n = 13$) rhesus macaques (*M.mulatta*) ranging in age from 3 to 10 years were evaluated in this study. In all groups, animals were assigned and evenly distributed based on age, sex, and MHC compatibility needs for each group and/or project.

**Multiplex bead Ab binding assay**

*Preparation of antigen-coated microspheres.* Antigens included recombinant gp120$_{C/ZM53}$, gp120$_{B/YU2}$ and gp120$_{A/MG505}$ from Immune Tech (New York, NY). To ensure sufficient coupling of peptides to the beads, all peptides used bore an N-terminal 6x Lys-Gly (KG)-linker. Peptides purchased from GenScript (Piscataway, NJ) included: clade C consensus V3 linear peptide (NNTRKSIRIGPGQTFYATGDIIG), cyclic V2$_{AE/92TH023}$ peptide (CSFNMTTELRDKKQKVHALFYKLDIVPIEDNTSSSEYRLINC), cV2$_{C/ZM109}$ peptide (CSFNITTDVKDRKQKVNATFYDLDIVPLSSSDNSSNSSLYRLISC), cV2$_{C/1086}$ peptide (CSFKATTELKDKKHKVHALFYKLDVVPLNGSSSSGEYRLINC), cV2$_{B/YU2}$ peptide (CSFNITTSI RDKVQKEYALFYNLDVVPIDNASYRLISC), cV2$_{A/MG505}$ peptide (CSFNMTTELRDKKQKVYSLFYR LDVIQINENQGNGSNNSNKEY RLINC), linear V2(AA166-185)$_{AE/A244}$ peptide (NMTTELRDKKQKVHALFYKL), linear V2(AA172-191)$_{AE/A244}$ peptide (RDKKQKVHALFYKLDIVPIE), and linear V2(AA190-209)$_{AE/A244}$ peptide (IEDNNDNSKYRLINCNTSVI). C5$_{C/ZM109}$ linear peptide (VEIKPLGIAPTEAKRRVVQREKR) was purchased from BioPeptide (San Diego, CA). V1V2-scaffold proteins bearing V1V2 domain inserts from a variety of HIV-1 strains and clades included V1V2-1FD6[24,55], V1V2-tags, and V1V2-gp70 reagents provided by B. Haynes and J. Peacock (Duke University) and V1V2$_{B/CaseA2}$-gp70 provided by A. Pinter (Rutgers University).

Antigens were covalently coupled to magnetic beads using a two-step carbodiimide reaction with the xMAP Antibody Coupling (AbC) Kit according to manufacturers' instructions (Luminex, Austin, TX). Carboxylated xMAP beads were coupled to 0.5 µg protein/million beads (V1V2$_{AE/A244}$-1FD6, V1V2$_{C/ZM109}$-1FD6 and V1V2$_{B/YU2}$-1FD6) or 1 µg protein/million beads (all peptides and V1V2$_{A/MG505}$-1FD6) or 4 µg protein/million beads (all gp120s, all V1V2-tags and -gp70, and BSA). The coupled beads were counted, diluted to a concentration of 500,000 beads/ml and stored at 4 °C for up to 1 month prior to use. The concentrations used on the beads were established on the basis of the reactivity with mAbs which were tested at concentrations ranging from 5 to $10 \times 10^{-6}$ µg/ml.

*Bead Ab binding assay.* First, a bead mixture was prepared by adding each bead type at 50 beads/µL in PBS-TB (PBS/0.1% BSA/0.02% Tween-20). In all, 50 µL/well (2500 beads of each bead type/well) were aliquoted from this mixture into black, clear bottom 96-well plates (Greiner Bio-One, Kremsmuenster, Austria). In total, 50 µL plasma samples were incubated with the bead mixture for 1 h at room temperature in the dark with shaking. Wells were washed twice with 100 µL/well PBS-TB and incubated with 100 µL/well of biotinylated anti-monkey IgG (2 µg/mL) (Rockland Immunochemicals, Pottstown, PA) or anti-human IgG (2 µg/mL) (Abcam, Cambridge, UK) for 30 min at room temperature in the dark with shaking. After washing twice with 100 µL/well PBS-TB, samples were incubated with 100 µL 1 µg/mL Streptavidin-PE (BioLegend, San Diego, CA) for 30 min at room temperature in the dark with shaking. Wells were then washed twice with 100 µL/well PBS-TB and beads measured for PE fluorescence using a Luminex FlexMAP3D device with xPONENT 4.2 software. Beads coupled to BSA served as negative controls. A cocktail of mAbs composed of multiple V2i (697, 830A, 1393A), V2p (CH58), V3 (3869), and C5 (670, 1331A) mAbs[19,57,59,60,92,93] was used as a positive control. The conditions of the assay were established so that the readings used were from the mid-point (most sensitive part) of the sigmoidal titration curve, rather than using saturating conditions. This allowed inter-experimental standardization.

NHP plasma were first screened for reactivity at a 1:200 dilution, and for titrations, dilutions ranged from 1:100 to 150,000. Samples were tested in duplicates and results shown as mean fluorescence intensity (MFI). Titration curves and scatter plots were generated in GraphPad Prism 7.03. AUC values were based on titration curves and divided by a denominator giving values between 0 and 15.

**Circular dichroism measurements.** Circular dichroism (CD) spectra were recorded on a Jasco J-1500 spectrometer equipped with a temperature controller using 1 mm length cells and a scan speed of 4 nm/min. The spectra were averaged over eight scans with the background subtracted according to the analogous experimental conditions. The peptides and proteins were prepared at 12.5–50 µM in 0.1x PBS (pH 7.4) and measured at 25 °C. Helical propensity titrations were conducted using 12.5–50 µM peptide in 50% v/v 2,2,2-trifluoroethanol (TFE) (Sigma-Aldrich, St. Louis, MO). CD data were fitted using the CD-FIT program (http://www.ruppweb.org/Xray/comp/cdfit.htm) and pie charts were generated based on the percentage of secondary structure (α-helix, random coil, or β-sheet) calculated by CD-FIT software.

**Rhesus macaque immunizations and challenges.** Experimental groups of adult rhesus macaques were infected with either a Tier 1 or Tier 2 SHIV, or immunized with one of three immunization regimens. The infected groups each consisted of ten animals that received four weekly intrarectal challenges with 5000 TCID$_{50}$ of Tier 2 clade C SHIV$_{ipd3N4}$[94] or three weekly intrarectal challenges with 8000 TCID$_{50}$ of Tier 1 clade C SHIV$_{1157ipEL-P}$[53]. Final blood draws occurred at 11 and 18 weeks after the last challenge dose, respectively. Twenty-five animals were

immunized using three different vaccine regimens (Table 1). One group of nine animals was co-immunized with *env-gag* DNA (36 μg of gp160$_{92TH028\_F8}$ and V1R SIV *gag* OPT) and 100 μg each of gp120 proteins from clade B MN and CRF01_AE A244 delivered intramuscularly (i.m.) ("DNA + gp120" group). Four immunizations were delivered at weeks 0, 4, 12, and 20[62]. The second immunized group consisting of 10 animals received a prime of the Simian Adenovirus 7 vectors (SAd7) carrying *gp150$_{C/1086}$* and *gag$_{SIVmac239}$* transgenes ($1 \times 10^{11}$ virus particles, i.m. and $0.5 \times 10^{11}$ virus particles, intranasally) at weeks 0 and 4 and a protein boost with soluble trimeric gp140$_{C/1086}$ and GBV-C E2 protein (25 μg each) at weeks 4 and 16, i.m.[53]. The third group of six animals was co-immunized with a total of 36 μg gp120$_{C/ZM53}$ DNA intradermally and three V1V2-scaffolds including V1V2$_{C/ZM53}$-2F5K, V1V2$_{C/ZM109}$-TTB, and V1V2$_{AE/A244}$-2J9C (50 μg each), i.m., at weeks 0, 8, 20[22,63]. All 25 immunized animals received their protein vaccines together with Adjuplex. Data shown for all three immunization groups are for blood specimens drawn 2 weeks after the last boost.

**C1q binding ELISA to measure complement activation**. C1q binding to V2- and gp120-specific Abs was measured by ELISA. Immulon 4HBX 96-well plates (Thermo Scientific, Waltham, MA) were coated with 2 μg/ml of the following Ags: cV2$_{C/1086}$, cV2$_{AE/92TH023}$, V1V2$_{C/ZM109}$-1FD6, V1V2$_{B/YU2}$-1FD6, gp120$_{C/ZM109}$ or gp120$_{B/YU2}$. Plates were incubated at 4 °C overnight, washed with 0.05% bovine serum albumin (BSA)/PBS, and blocked in PBS/7.5% BSA for 1.5 h at room temperature. Plasma samples were heat inactivated at 56 °C for 30 min, titrated at 2-fold dilutions ranging from 50x to 400x and added to the plate. After 1.5 h incubation at room temperature, plates were washed and incubated with 10 μg/ml of C1q (Sigma-Aldrich, St. Louis, MO) for 1.5 h at room temperature with shaking at 500 rpm. The C1q was diluted in PBS/ 0.5% BSA/0.05% Tween 20. After washing, bound Abs were detected with 10 μg/ml mouse monoclonal IgG1 anti-human C1q-HRP (Santa Cruz, Dallas, TX) diluted in PBS/0.5% BSA/0.05% Tween 20 and incubated for 1 h at room temperature with shaking at 500 rpm. After washing, plates were developed with 3,3′,5,5′-tetramethylbenzidine (TMB) substrate (Thermo Scientific, Waltham, MA) followed by 1 N hydrochloric acid and read at 450 nm on a BioTek PowerWave HT plate reader (Winooski, VT) with Gen5 software version 3.08. Assays were standardized with positive controls (mAb CAP228-16H for V2p Ags, mAb 830 A for V2i, and gp120 Ags), and a negative control anti-anthrax PA-specific mAb 3685 was included with each assay. Background values from pre-bleed specimens were subtracted from the values obtained with specimens drawn 2 weeks after the last immunization or 18/11 weeks after infection, and the area under the curve (AUC) was determined.

**ADCP**. Antibody-dependent cellular phagocytosis (ADCP) was performed by measuring the uptake of HIV Env Ag-coated microspheres by the monocytic THP-1 cell line (ATCC, TIB-202). For this, 5 μg of Ag (cV2$_{C/1086}$, cV2$_{AE/92TH023}$, V1V2$_{C/ZM109}$-1FD6 or V1V2$_{B/YU2}$-1FD6) was biotinylated using EZ-Link Sulfo-NHS-LC-biotin (Thermo Scientific, Waltham, MA) and then conjugated to 1 μm fluorescent neutravidin beads (Thermo Scientific, Waltham, MA) according to the manufacturer's instructions. Conjugated beads were washed and resuspended in PBS/ 0.1% BSA. A total of $1.8 \times 10^6$ beads was aliquoted per well in round bottom 96-well plates. Plasma was titrated using five-fold dilutions and added to the beads, followed by incubation for 2 h at 37 °C and a washing step. A total of $2.5 \times 10^4$ THP-1 cells was added to each well and incubated overnight at 37 °C. Assays were standardized with positive controls (mAb CAP228-16H for V2p Ags, mAb 830 A for V2i, and gp120 Ags), and a negative control anti-anthrax PA-specific mAb 3685 was included with each assay. Phagocytosis was measured by flow cytometry using a BD LSR II Fortessa equipped with an HTS plate reader. Data were acquired with the BD FACSDiva Software version 8.0.2 (Franklin Lakes, NJ) and analyzed by FCS Express 7 Flow Research Edition (De Novo Software; Pasadena, CA). The gating strategy is supplied in Supplementary Fig. S9. ADCP scores were calculated by multiplying the percentage of bead-positive cells by mean fluorescent intensity (MFI) and dividing by a denominator giving values between 0 and 10. AUC values were calculated from the titration curves as described for C1q binding.

**Software scripts and visualization**. Normalized heatmaps were generated using the complexheatmap and tidyverse packages in program R x64 version 4.0.2 and RStudio version 1.3.959[95,96]. Normalizations were done for each assay type/Ag combination (parameter). Principal component analyses were done with factoextra and tidyverse packages using prcomp function based on singular value decomposition in R and RStudio.

For generating the correlogram, the corrplot and RColorBrewer packages in R and RStudio using hierarchical clustering (hclust) were used. Dendrograms were calculated using the dendPlot function and hclust method, or as implemented in the complexheatmap packages in R. Annotations were done using thecomplexheatmap package.

The correlation network diagram was generated in undirected mode in R and RStudio using ggraph, igraph and tidyverse packages, with clustering based on the graphot layout. Edges are weighted according to *r* values. Edges are only shown if *p* < 0.05, and nodes without edges were removed. Nodes are sized according to the *r* values of the connecting edges.

**Data analysis**. Unless otherwise noted, data analyses were performed with Microsoft Excel 2013 (Redmond, WA), GraphPad Prism 7.03 (San Diego, CA), R x64 version 4.0.2 (Indianapolis, IN), and R studio version 1.3.959 (Boston, MA).

**Statistical analyses**. D'Agostino and Pearson normality tests were performed with the software GraphPad Prism 7.03 (San Diego, CA). The statistical comparison of groups was done using the two-tailed Wilcoxon matched pairs signed rank test (mAb analyses). Multiparameter pairwise correlation analyses were done using Pearson's correlation as data were in a linear range, following bivariate distribution. Correlation coefficients *r* and *p* values were calculated in GraphPad Prism and a *p* value < 0.05 was considered significant. Multiplicity adjustments for *p* values were performed with the Benjamini–Hochberg method in R and R Studio using the data.table and tidyverse packages[95,96].

**Reporting summary**. Further information on research design is available in the Nature Research Reporting Summary linked to this article.

## Data availability
Luminex binding, C1q and ADCP data generated in this study are provided in the Supplementary Information and the Source Data file. Source data are provided with this paper.

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

## Acknowledgements

This work was supported in part with funds from the National Institute of Allergy and Infectious Diseases, P01 AI100151 (PIs: X.-P.K. and S.Z.-P.), R01 AI145655 (PI: X.-P.K.), R01 AI112546 (PI: M.K.Gorny), P51 OD011092 (N.L.H.), P01 AI078064 (N.L.H.), U42 OD010426 (N.L.H.), R44 AI091546 (J.A.), R01 AI122953-05 (PI: R.D.), and R01 AI104387 (L.M.), as well as the South African Medical Research Council (L.M.) and the Department of Medicine, Icahn School of Medicine at Mount Sinai (S.Z.-P.).

## Author contributions

S.W., J.A., N.L.H., A.J.H., M.K.G., and S.Z-P. conceived and designed experiments. S.W., V.I., R.P., D.C.M., and P.B. performed the experiments. X.J. and C.L. produced protein immunogens. L.M. provided the mAbs CAP228-19F, -3D.1, and -16H. R.P. performed the circular dichroism experiments. S.W., R.P., X.-P.K., A.J.H., R.D., and S.Z.-P. analyzed and interpreted data. S.W. and S.Z.-P. wrote the manuscript; and all authors read, edited, and approved the manuscript.

## Competing interests

The authors declare no competing interests.
