## [Peer Review File · Nature Communications]

Differential V2-directed Antibody Responses in Non-human Primates Infected with SHIVs or Immunized with Diverse HIV VaccinesReviewers' Comments:

Reviewer #1:

Remarks to the Author:

In this study the authors have characterized V2 directed "V2p" and "V2i" Abs based on the antigens they recognize across different clades. V2p class of mAbs bind clade specific cyclic V2 peptides; while the V2i class of mAbs recognize scaffolded V1V2 fusion proteins. It was interesting to look at the comparison between the immune responses elicited by Tier-1 SHIV1157ipEL-p and Tier-2 SHIV1157ipd3N4. The authors found Tier-1 infected macaques to elicit significantly higher (V2i+V2p) directed Abs than the Tier-2 infected macaques. However, these responses were significantly weak compared to responses from immunized macaques. Immunization regimen containing V1V2-scaffold immunogens elicited strongest V2i directed Abs compared to gp120 and gp140 immunization regimens tested in the study. Overall, the study is interesting and the analyses are done carefully. However, the major issue is with the significance of the results. We do not know what do these differences mean in terms of function and possibly protection? It is very important to look at the functions of the V2i and V2p kind of Abs elicited by the various immunization regimens tested in the study. Following are the additional concerns.

1. Fig 2c claims difference in secondary structure content of the V2p and V2i Ab directed antigens. However, the CD data in Fig 2a, 2b shows data for only cV21086, cV292TH023 and V1V2-1086-tags which are recognized by V2p Abs. It will be useful to extend the data across different V1V2 antigen forms recognized by V2i mAbs mentioned in the study and other cyclic V2 and V1V2 tags. Ideally, the authors should look at various forms of the same envelope sequence and possibly extend this to more than one strain.

2. Fig 1a shows the reactivity of various V2 targeting antigens to V2 directed mAbs. Specificity of the mAbs has not been monitored with corresponding variants e.g. there is data showing V2p Abs binding to cV21086, cV292TH023, but there isn't any binding data for corresponding V1V2 tags or scaffold fusion protein versions of 1086 and 92TH023; and vice-versa for many other scaffold fusion proteins listed in the study. This is important to select antigens which define V2p and V2i Abs.

Reviewer #2:

Remarks to the Author:

Antibody responses against the HIV-1 envelope V1V2 region have been a subject of interest from a vaccine perspective because of their association with reduced acquisition in RV144 vaccinated individuals. Multiple types of V2-directed antibodies have been described that recognize linear epitopes (V2p), discontinuous epitopes that overlap with the a4b7 integrin binding motif (V2i), quaternary epitopes involving V2 glycans (V2q), and trimer dependent V2 epitopes (Vqt). While V2p and V2i are generally not able to neutralize HIV-1 variants, V2q and V2qt antibodies have been associated with broad neutralization. However, V2p and V2i antibodies can mediate Fc effector functions and can be elicited by vaccination in multiple animal models and human vaccinees.

In this interesting paper, the authors describe an approach to use envelope antigens that present various forms of the HIV-1 envelope V1V2 region to characterize polyclonal serum/plasma antibody responses for V2 specificity. They utilize well-characterized monoclonal antibodies and various multiplexed envelope antigen forms to validate their ability to distinguish between V2p and V2i antibodies. They then apply this approach to characterize and compare the polyclonal V2 directed antibody responses in 25 immunized and 20 SHIV-infected rhesus macaques.

The paper shows that three different immunization regimens elicited V2p and V2i antibodies in rhesus macaques, with one of the regimens eliciting higher levels of V2i antibodies than the other two. The paper also shows that infection with a tier 1 or tier 2 SHIV1157 variant elicits very weak V2p or V2i antibodies. The authors conclude that immunization and infection elicit significantly different patterns

of V1V2-specific antibodies. Overall, the approach and the findings are very interesting and provide an important focus away from bnAbs on antibodies that have been associated with reduced acquisition in vaccinated humans and are readily elicited by different types of vaccines and immunogens. A better understanding of how to elicit these potentially protective antibodies would help to advance vaccine development. However, there are some concerns with the experimental design and interpretation of the data, mainly that envelope sequence variation is not incorporated into the study, that if addressed would elevate the significance and impact of the study.

Comments:

1. The authors should address the effect of time of sampling between the SHIV infections and immunizations. The antibodies were examined at 37/38 weeks post-infection, at the time of necropsy. Are earlier samples available from the SHIV infected macaques? How would mutation of the V1V2 region of the SHIV in vivo influence the presence of antibodies that would cross-react with the antigen panel?
2. The authors should address the possibility that sequence variation across the many antigens and the SHIVs could influence the results. For example, the contact sites for CH58 and CH59 and the CAP228-16H mAbs are known and SHIV1157 lacks some of the contact residues for CH58 and CH59, including 173H, so SHIV infection would not be expected to induce antibodies with those reactivities. How much of the lack of reactivity with the SHIV-infected plasma could be explained by the SHIV mismatching with key residues needed for recognition of the antigens? If those antigen forms were constructed with SHIV autologous sequences, could V2 antibodies be detected?
3. Similarly, the vaccine immunogens consist of envelopes of clades B, CRF01, and C, and the sequence differences should be considered in reactivity patterns for the immunized animals. Why are there no C5 directed antibodies detected in the Sad7 plus gp140 immunized animals? Wouldn't the 1086.C gp150/140 vaccine regimen be expected to elicit C5 antibodies? If an immunogen-matched C5 antigen is used, are antibodies detected in those animals?
4. Ideally, the different types of antigens would be matched so that lack of binding could be attributed to the conformation of the epitope, excluding sequence mismatches. For example, there would be gp120, V2 peptide, V1V2 tag, V1V2-gp70, and V1V2-1FD6 all based on the same envelope variant.
5. Was there protection in any of the vaccination regimens, and was it associated with the presence or levels of the V2p/V2i antibodies?
6. The paper would be strengthened even more by analyzing Fc effector functions and associating those with the presence of the different V2 specificities.

Reviewer #3:

Remarks to the Author:

The manuscript by Weiss et al. titled "Differential V2-directed antibody responses in non-human primates 2 infected with SHIVs or immunized with diverse HIV vaccines" described the application of a panel of HIV Env-based antigens including the antigens presenting various V1V2 epitopes to profile antibody responses induced by SHIV infections and immunizations in non-human primates, respectively. The authors found that the sera from immunized animals all displayed moderate to strong binding to V2P (linear V2 peptide) or V2i (conformation-dependent epitope) antigens, whereas the sera from infected animals showed very limited binding to such antigens. The delineation of polyclonal serum antigen-specific antibody response is important for vaccine design and improvement. The antigen panel described in this study would be valuable reagents for analyzing V1V2 antibody response induced by vaccination and viral infections.

Major point:

- 1) It has been demonstrated that the HIV V1V2 region is the target of broadly neutralizing antibodies, as well as non-neutralizing antibodies possessing effector functions such as ADCC/ADCP/ADCVI that may facilitate killing of infected cells. The representative V2 antibodies such as 830A and 697-30D used as reference antibodies (Figure 1) in this study were isolated from HIV-1 infected individuals. However, the finding in this manuscript that viral infections (represented by SHIV herein) induced very weak or no antibody response targeting the V1V2 region is not consistent with the work presented previously by this group of scientists and others that HIV V1V2 is highly immunogenic. Thus, the authors are encouraged to clarify the observation of the current study in the context of the findings of the field on the immunogenicity of HIV-1 V1V2 region. It is not clear if the SHIV infections described in this study are atypical viral infections. It would be helpful if the authors could use this panel of antigens to profile the sera from HIV-1 infected individuals in addition to the SHIV infected animals.
- 2) The rationale for choosing the antigens in Fig. 3 to profile antibody response needs to be clarified. Why there is no V2 peptide or V1V2 mini protein (V1V2-tag or V1V2-gp70) derived from the SHIV strains used to infect animals, while the V1V2 antigens are more related to the immunogens used to immunize animals in the immunization groups? Will this apparent antigen bias lead to the conclusion that V2P and V2i response is weak for the SHIV infected animals?
- 3) It is important for the authors to show the antigenicity of the immunogens described in this study Table 1. how do they react with these V2 mAbs in Fig. 1? Some immunogens are listed in Fig. 1 as antigens. However, the other immunogens such as gp120s of MN and A244, gp140 C-1086 (is this in the uncleaved gp140 form or native-like SOSIP form?)
- 4) It is suggested that the immunizations described in this study induced antibody responses with V2-directed ADCC/ADCP capacity, while the SHIV infection had such response much weaker or no such responses. To support this notion, it is important to show the V2-directed ADCC/ADCP capacities of sera of animals from the immunized animals against the SHIV strains (especially the tier2 SHIV strain) in this study, in comparison with the sera from the SHIV infected animals.

Minor points:

- 1) It is not clear in the main text when the animal sera were collected for the SHIV infected animals (e.g. how many weeks post the last virus inoculation) and immunized animals (e.g. how many weeks from the last immunization). Such information is included in the supplementary figure legends. It will be helpful to specify this in the method part.
- 2) In Fig.3 the animal plasma binding activity is presented as MFI, in Fig. S2, it is presented as deltaMFI. Please use consistent label. The plasma dilution factor (1:200?) should be shown in the figure legend. In addition, the figure legend of Fig.3 reads "Intensity of the reactivity is shown as the mean MFI calculated from experiments normalized on the basis of the reactivity of the mAb pool in each experiment from which background (PBS-TB) was subtracted". The description of MFI data process in this figure legend and in the method section is not clear. The authors are encouraged to clarify how the MFI data is processed, how the normalization is done, why and how the reactivity of the mAb pool is normalized.
- 3) Line 384-385 reads "No single amino acid or combinations of amino acids characterizes the reactivity of this mAb", what does this mean?
- 4) Line 400-401 reads "Peptide array data show that Abs reactive with various linear V2 peptides were induced in both the RV144 and VAX003 human vaccine trials, but not in VAX004". Please remind the reader what are the VAX003 and VAX004 trials. Are they related to any of the immunization schemes described in this study? This will be helpful for people to understand how the V2 linear peptide reactivity related to these trials.
- 5) Line 426-427 reads "This indicates that vaccines can induce different and more potent V2 Ab responses compared to infection". In terms of "more potent", what kind of potency is referred to? Is it referred to neutralization, ADCC, or just antigen binding?

Response to Reviewers' Comments and Additional Information about Highlighted Changes

As requested, we are providing this response to the reviews that were sent to us after the first submission of this paper to *Nature Communications* in late 2019. We have addressed all of the comments of all three reviewers and the specific changes are noted below in **bold**.

Please note we have **highlighted and underlined** text in the manuscript that was changed/added in response to the Reviewers' comments. Other additions to the text, made by the authors to clarify or extend the data are **highlighted but not underlined**. Areas of tables and figures that were edited as per reviewers' request, edited for clarity, or added are **highlighted**. References that were added relevant to the additions and changes, as well as references published since the initial submission are also **highlighted**.

Reviewer 1: "In this study the authors have characterized V2 directed 'V2p' and 'V2i' Abs based on the antigens they recognize across different clades. V2p class of mAbs bind clade specific cyclic V2 peptides...Overall, the study is interesting and the analyses are done carefully."

Overall: It is very important to look at the functions of the V2i and V2p kind of Abs elicited by the various immunization regimens tested in the study.

Response. New experiments measuring the ADCC, ADCP and complement activating functions of V2 Abs were performed and have been added. The data are found in newly added **Figures 4, 5 and 6, Supplementary Figures S4, S5, S6, and S7 and in the legends describing these figures**. These are described in the text and are highlighted and underlined, e. g., **lines 44-50, 97-98, 108-11, 238-80, 360-2, 421-8, 434-8**). Statistical analyses of all data, including the functional data have also been added (**lines 281-294, 304-25, 515-50**).

Comment 1a. Figures 2a and 2b show data for only cV2₁₀₈₆, cV2_{92TH023} and V1V2₁₀₈₆-tags which are recognized by V2p Abs. It will be useful to extend the data across different V1V2 antigen forms recognized by V2i mAbs mentioned in the study and other cyclic V2 and V1V2 tags.

Response: As requested by the reviewer, **Figure 2a** now includes new data for the CD spectra of cV2_{A/MG505}, cV2_{C/ZM109}, cV2_{B/YU2}, V1V2_{AE/A244}-tags, and V1V2_{B/CaseA2}-tags. In addition, **Figure 2b** has added data describing the configuration of the V2 and V1V2 segments of these reagents. These data now provide comparisons for the configurations of 5 cyclic peptides and for 3 V1V2-tags antigens, and in addition provide a comparison for cV2_{C/1086} vs V1V2_{C/1086}-tags and for cV2_{AE/92TH023} vs. V1V2_{AE/A244}-tags (which have the same V2 sequences). The data are described in the text on **lines 154-158**.

We note that the reviewer had asked for CD spectra of V1V2-scaffold proteins, but since the V1V2 segment is only a small portion of such a molecule, the CD spectra would provide no information about the conformation of the V1V2 segment.

Comment 1b. Ideally, the authors should look at various forms of the same envelope sequence and possibly extend this to more than one strain.

Response: See response in Comment 2.

Comment 2. Specificity of the mAbs has not been monitored with corresponding variants e.g. there is data showing V2p Abs binding to cV2₁₀₈₆, cV2_{92TH023}, but there isn't any binding data for corresponding V1V2 tags or scaffold fusion protein versions of 1086 and 92TH023; and vice-versa for many other

scaffold fusion proteins listed in the study. This is important to select antigens which define V2p and V2i Abs.

Response: We have now added Luminex data (as well as CD spectra) for additional cyclic V2 peptides to provide comparisons between “corresponding variants”. Thus, **Figure 1a** and **3** now include new data for 6 new V2 peptides. As per the reviewer’s request, we now show data for V2 reagents with identical V2 segments:

V1V2_{C/ZM109}-1FD6 (scaffold protein) and cV2_{C/ZM109} (cyclic peptide)

V1V2_{B/YU2}-1FD6 and cV2_{B/YU2}

V1V2_{A/MG505}-1FD6 and cV2_{A/MG505}

V1V2_{AE/A244}-1FD6 and cV2_{AE/92TH023} (A244 and 92TH023 have the same V2p sequence)

Also, data now include:

A/MG505 tested as a gp120 protein, a cV2 peptide, and a 1FD6 scaffold; AE/92TH023

AE/A244 (with identical in V2 sequences) as a cV2 peptide, a tags peptide and a 1FD6 scaffold

C/ZM109 as a cV2 peptide and a 1FD6 scaffold

C/1086 tested as a cyclic peptide and a tags peptide.

Additional data with the newly added cV2 and V1V2-1FD6 reagents now also appear in the titration curves in **Supplementary Figures S1, S3a, S3b, and S3c** and in the text of the Results section (**lines 115-9, 122-121**) and Methods (**lines 451-8**).

Data with linear peptides spanning different regions of V2 have also been added and these are described on **lines 115-9, 122-9, 212-6, 453-60**.

Reviewer 2: “The authors conclude that immunization and infection elicit significantly different patterns of V1V2-specific antibodies. The approach and the findings are very interesting and provide an important focus away from bnAbs on antibodies that have been associated with reduced acquisition in vaccinated humans and are readily elicited by different types of vaccines and immunogens.”

Overall: There are some concerns, mainly that envelope sequence variation is not incorporated into the study, that if addressed would elevate the significance and impact of the study.

Response: The data and reagents chosen for analysis were indeed selected to take into account the sequence variation; despite this variation, broad cross-reactivity was observed. To clarify this, we have added back a version of **Supplementary Table S1** (expanded from the one included in the original submission of 2019) which shows the sequences of the V2p epitope from strains from clades A, B, C and AE as well as from the two clade C SHIVs used in this study (see also **lines 115-119, 122-9, 209-11, 365-8**). Text was already in the manuscript describing the cross-reactive V2 Abs of the immunized animals, and now text has been added concerning the similarities and differences among the V2p sequences (**lines 212-6**) and the broad cross-reactivity of the V2i and V2p mAb (**lines 122-129**). Examples include:

–(a) strong cross-reactivity as shown in **Figure 1a** by mAb V2p CAP228-16H with reagents from three clade C strains (ZM53, 1086, 97ZA012), two clade A strains (MG505 and 92RW020), two clade AE strains (92TH023 and A244) and one clade B strain (CaseA2). As shown in **Supplementary Table S1** and **lines 118-9, 122-9**, strains C/1086 and A/MG505 differ in sequence by 32% (6 of 19 positions in V2) and yet have similar and strong reactivity with CAP228-16H;

-b) strong cross-reactivity was also noted using the plasma from NHPs immunized with the DNA + V1V2-scaffold regimen and V2 reagents from AE/92TH023/A244, C/1086, C/ZM109, B/YU2 and

A/MG505. Again, strains C/1086 and A/MG505 differ by 32% and yet the immune plasma react to these reagents similarly (**Figure 3 and lines 122-9**).

Comment 1a. The authors should address the effect of time of sampling between the SHIV infections and immunizations. The antibodies were examined post-infection at the time of necropsy.

Response: The reviewer is correct that the plasma tested were from immunized animals two weeks after the last immunization whereas the plasma tested from infected animals were derived at 11 or 18 weeks after infection. We note that the response of the infected and immunized animals were comparable against the three gp120 proteins from clades A, B and C as well as against a V3 peptide that was heterologous to the infecting strains. This suggests that the responses were comparable in infected and immunized NHPs to immunodominant epitopes which suggests that if there were a substantial Ab response to V2 epitopes, it would not be expected to decline more rapidly than the Abs to V3 and gp120 epitopes. Text to this effect has been added (**lines 182-4, 202-11, S20-1**).

Comment 1b. How would mutation of the V1V2 region of the SHIV in vivo influence the presence of antibodies that would cross-react with the antigen panel?

Response: Given the similar reactivity between cV2_{C/1086} and cV2_{C/ZM109}, and (**Figure 1a**) which differ from one another in 8 of 19 amino acid residues (**Supplemental Table S1**), it is unlikely that mutations that might occur in the V2 region after 18 weeks of SHIV infection would account for a loss or gain of cross-reactive Abs. This reviewer's comment recapitulates his/her concerns about cross-reactivity which were addressed originally and now have been expanded as noted above and below. The issues of mutation and cross-reactivity with between V2 sequences is again addressed in **lines 202-11**.

Comment 2. The authors should address the possibility that sequence variation across the many antigens and the SHIVs could influence the results.

Response: This issue of sequence variation vs. cross-reactivity is addressed above in the response to the comments of Reviewer 2 under the headings of Overall and Comment 1b, and new text addressing this issue has been added (**lines 122-9, 202-11**). In addition to descriptions and discussions of cross-reactive response in several other areas of the manuscript, we note again that we have now added the V2 sequences of the SHIV Tier 1 and Tier 2 viruses used for infection to **Supplementary Table S1**. Notably, these differ in mutated residues from the conserved residues in far fewer positions than, for example, the V2 of C/ZM109 (4, 2 and 7 residues, respectively).

Comment 3. Why are there no C5 directed antibodies detected in the SAd7 plus gp140 immunized animals? Wouldn't the 1086.C gp150/140 vaccine regimen be expected to elicit C5 antibodies? If an immunogen-matched C5 antigen is used, are antibodies detected in those animals?

Response: It is unlikely that the absence of detectable C5 Abs is due to a failure to react with the C5 peptide because (a) intact gp140_{SF162} was previously shown incapable of inducing C5 Abs (Srivastava IK et al., J. Virol., 2003), C5 Abs have been shown to be cross-clade reactive (Nyambi P et al., J. Virol., 1998; Chen Y, et al., J. Virol., 2013) and given the conservation of C5 ("conserved" region 5), it is probable that if the animals immunized with a clade C gp140 could mount C5 Abs, their Abs would react with the C5_{consC} peptide. New text concerning this has been added to **lines 197-9**.

Comment 4. Ideally, the different types of antigens would be matched so that lack of binding could be attributed to the conformation of the epitope, excluding sequence mismatches. For example, there would be gp120, V2 peptide, V1V2 tag, V1V2-gp70, and V1V2-1FD6 all based on the same envelope variant.

Response: A similar request was made by Reviewer 1 and as described in the detailed response to Reviewer 1's Comment 2, we have now added new data in several graphs and tables and in the text concerning several new reagents that are "matched" forms of various V1V2 and V2 antigens.

Comment 5. Was there protection in any of the vaccination regimens, and was it associated with the presence or levels of the V2p/V2i antibodies?

Response: (a) In the Discussion of the revised manuscript submitted last month (**lines 389-98**), we had added that protection was associated with the presence of V2 Abs for the DNA + gp120 group. These data were published during this last year and are newly cited (**reference #62**). (c) No challenge study has yet been performed with the DNA + V1V2-scaffold group.

Comment 6. The paper would be strengthened even more by analyzing Fc effector functions and associating those with the presence of the different V2 specificities.

Response: As noted in the response to Reviewer 1's overall comments, new experiments measuring the functions of V2 Abs were performed and have been added. The data are found in newly added figures: **Figures 4, 5 and 6** and **Supplementary Figures S4, S5, S6, and S7**.

Reviewer 3. "The delineation of polyclonal serum antigen-specific antibody response is important for vaccine design and improvement. The antigen panel described in this study would be valuable reagents for analyzing V1V2 antibody response induced by vaccination and viral infections."

Comment 1a. The finding in this manuscript that viral infections (represented by SHIV herein) induced very weak or no antibody response targeting the V1V2 region is not consistent with the work presented previously by this group of scientists and others that HIV V1V2 is highly immunogenic. Thus, the authors are encouraged to clarify the observation of the current study in the context of the findings of the field on the immunogenicity of HIV-1 V1V2 region.

Response: This reviewer has misinterpreted previously published data. The V1V2 region is not "highly immunogenic". In fact, we recently published a paper (Liu et al., *Virology* 2019) showing that there were "broad variations in levels of anti-V2 Abs, and 6 of the 79 plasma samples tested longitudinally displayed substantial deficiency of V2 Abs". Text and citations have been added to clarify this point (**lines 356-8** and **reference 83**).

Comment 1b. It would be helpful if the authors could use this panel of antigens to profile the sera from HIV-1 infected individuals in addition to the SHIV infected animals.

Response: We agree with the reviewer that the Ab profile of immunized vs. HIV-infected humans is important. It is, in fact, the subject of a paper now in preparation. The data are extensive and we feel it would not be advisable to incorporate so much data in a single manuscript that would be overly long and would report two related but distinct stories.

Comment 2. Why are there no V2 peptides and/or V1V2 mini proteins (V1V2-tag or V1V2-gp70) derived from the SHIV strains used to infect animals, while the V1V2 antigens are more related to the immunogens used to immunize animals in the immunization groups? Will this apparent antigen bias lead to the conclusion that V2p and V2i responses are weak for the SHIV infected animals?

Response: This concern has already been addressed in terms of sequence variation and cross-reactivity between strains, clades and HIV vs. SHIV sequences (see Responses to the comments of Reviewer 2 under the headings of Overall and Comments 1b and 2, above). As noted, new text addressing this issue has been added in several areas of the manuscript and in **Supplementary Table S1**.

Comment 3. It is important for the authors to show the antigenicity of the immunogens described in this study. How do they react with the V2 mAbs in Fig. 1?

Response: The antigenicity of several of the V1V2-scaffold immunogens used in the studies described in this manuscript was previously published and is cited at several points in the manuscript (reference #22). This information has been added to the text (**lines 358-60**). In addition, it is noteworthy that antigens used for immunization included various forms of gp120 or gp140 from strains C/92TH028, B/MN, AE/A244, C/1086, C/ZM53, and/or C/ZM109. The antigenicity/reactivity of the Env, peptides, and/or V1V2-scaffold proteins from all of these strains (with the exception of B/MN) was tested with mAbs, and the results are in **Figure 1**. The author also queried if the gp140 used in the SAd7 + gp140 group was trimeric. This has now been clarified (**line 509**).

Comment 4. It is important to show the V2-directed ADCC/ADCP capacities of sera of animals from the immunized animals against the SHIV strains (especially the tier 2 SHIV strain) in this study, in comparison with the sera from the SHIV infected animals.

Response: This reviewer again raises the issue of cross-reactivity, here in terms of functional ADCC/ADCP Ab activities. The issue of cross-reactivity has already been addressed above, is now addressed at several new points in the manuscript (as noted above). Of note, in **Supplementary Table S1** we show the V2 sequences of the Tier 1 and Tier 2 SHIVs used to infect the monkeys. It is notable that the V2 sequence of the Tier 2 SHIV has only two substitutions at the most common V2 residues, and neither of these are at known Ab contact points. This is now discussed in greater detail on **lines 205-11**.

In addition, we note again the comparable reactivity of the SHIV-infected and immunized NHPs to three gp120s from heterologous clades and strains as well as a V3 peptide. All of these points have now been expanded in the text as noted above. For all of these reasons, the data are consistent with cross-reactive Ab responses by both infected and immunized NHPs and we feel that it precludes the need for more data using SHIV reagents.

Minor points:

Responses: (1) The times of collection of plasma from SHIV infected and immunized animals have been clarified. (2) MFI data from all experiments is consistently shown as Δ MFI (**Figures 3, 4, S1, S2, and S3**). The dilution of the plasma used in Figure 3 is now shown in the legend as 1:200 (**line 880**). (3) We explain that normalization of Luminex experimental data was performed by means of using the same pool of mAbs as a positive control in every experiment and that the results of experiments performed on different days was normalized using the results from the positive mAb pool. This has now been clarified as requested (**lines 880-2, 888-91, and S26-8**). (4) Text discussing **Supplementary Table S1** has been added (**lines 115-9, 122-1, 205-9**). (5) Mention of VAX003 and VAX004 has been eliminated because they are irrelevant to this study. (6) The term “more potent V2 Ab responses” has been replaced with “higher levels of binding Abs”.

Additional information about highlighted changes (which are not underlined) that were added by the authors

--A new author, Ralf Duerr, Ph. D., and his affiliation have been added as he participated extensively in the providing statistical expertise and help in performing the newly added statistical analyses.

--**Figure 1b** has been added to show the differential reactivities of V2p and V2i monoclonal antibodies (mAbs) with cyclic V2 peptides vs. the V1V2-1FD6 scaffold protein. These data are now described on lines **141-8***.

--Several new and newly published references have been added (highlighted in the Reference section and in the superscripts in the text) and discussed when relevant in the text, e.g., **lines 60-62, 414-5**, etc.

Reviewers' Comments:

Reviewer #1:

Remarks to the Author:

The authors have addressed all my concerns satisfactorily.

Reviewer #2:

Remarks to the Author:

The authors have done a thorough job of revising the manuscript. They added data and experiments, and clarified some areas as requested by the previous reviewers. This has resulted in a stronger, more compelling manuscript. I have only a few minor comments and suggestions.

Figure 1:

Could the authors provide a clearer explanation of Fig. 1b – exactly which mAbs and antigens were tested in each panel?

Could the authors include a grouping of binding to matched antigens (i.e. MG505 as gp120, cV2, and 1FD6 antigens). This could even be supplementary.

Figure 2:

The authors might consider matching the colors in parts b and c.

Figure 3:

As for Fig. 1, the authors could indicate binding to matched antigens by grouping them.

Text:

Line 159 – should have Wibmer et al.?

Line 347 – remove parentheses.

Reviewer #3:

Remarks to the Author:

The revised manuscript has addressed the cross-reactivity of the immune sera by adding more antigens with matched V1V2 sequences. Overall, the authors have made a good point that the immunization regimen DNA+V1V2-scaffold has elicited higher titers of antibodies conferring complement deposition (C1q binding) and ADCP measured by V2 antigens including cV2 peptides and V1V2 scaffold than other immunization regimens (DNA+ gp120 or SAd7+gp140) involving larger HIV envelope molecules as immunogen. Such antigens could be utilized for detecting V2-specific antibody responses that may confer the above stated functions. The manuscript is quite focused on the elicitation of the V2-specific antibody responses, which would be more balanced by discussing the overall biological beneficial effect of immunization regimens and presenting data accordingly.

Major points

(1). The authors had provided new data of ADCP and C1q binding functional assay, using cV2 peptide or V1V1 scaffold as antigen. For HIV specific antibody response to confer ADCC/ADCP function, the antibodies have to bind the whole HIV envelope glycoproteins presented on the infected cells. It would be important for the authors to provide data using the whole gp140 trimer as antigen in such assay,

and also using infected cells as target cell for ADCC function. In reference 62, the difference of ADCC between the anti-Env and anti-V2 immunization regimen is not significant. It may be helpful to assess V2-specific ADCC activity in this assay.

(2). Adding control V2-specific mAbs in the assays in Fig. 4 could improve the data clarity. It is surprising that all the immune sera bind cV2 peptides, but the sera from either the DNA+gp120 or sAd7+ gp140 immunized animals displayed no C1q or ADCP activity when cV2 peptides were used as antigens. It is not clear if this observation is caused by the different isotypes of the immune serum antibodies or the peptide antigen could not mediate C1q binding, as the authors discussed. A control V2-specific monoclonal antibody such as CAP22816H (V2P) and 830A (V2i) would help to validate the assay.

(3). Regarding the antigenicity and immunogenicity of C5 region, there has been description in the literature that the exposure of C5 region has been dampened in trimeric gp140 context (J Virol. 74(12):5716-25), which may help with the discussion.

(4). Although it is encouraging that the DNA+V1V2-scaffold regimen has elicited higher V2i antibody response than the other regimens, the authors should discuss the limitation of this regimen. Indeed, as reference 62 reported, the ADCC, ADCP, and ADCVI scores of the animals from the anti-V2 regimen were not better than the anti-Env group. The authors stated in lines 421-422 that "In the data presented, only the V2-directed immunization regimen induced Abs with the Fc-mediated effector functions of ADCP and complement activation". This statement is not consistent with the context of the literature including reference 62. Therefore, this sentence could be revised.

Minor point

(1). There are numerous boxes with bold lines in Figures 1 and 3, which block other areas of the figure and could be removed to improve the clarity of the figures.

(2). Figure 5b, please explain what are Dim 2 and Dim 1, and how they have been generated? Either the method part or the figure legend provided sufficient information.

RESPONSE TO REVIEWER COMMENTS

Please note that all changes made in response to the Reviewers' most recent comments are highlighted in the text in magenta, and that, as a result of the addition of new supplementary figures, as requested, the designation for these throughout the text has been changed. Changes to the figures and supplementary figures have been shaded with pale purple boxes.

Reviewer #1 . The authors have addressed all my concerns satisfactorily.

Reviewer #2. The authors have done a thorough job of revising the manuscript. They added data and experiments, and clarified some areas as requested by the previous reviewers. This has resulted in a stronger, more compelling manuscript. I have only a few minor comments and suggestions.

Critique_ Figure 1: Could the authors provide a clearer explanation of Fig. 1b – exactly which mAbs and antigens were tested in each panel?

Response: As per the reviewers' request, the mAbs used in each of the graphs of Figure 1b have now been delineated in the legend for this figure (lines 960-3). In addition, color-coding of antigens used in Figure 1b has been added to show which antigens were used.

Critique_ Could the authors include a grouping of binding to matched antigens (i.e. MG505 as gp120, cV2, and 1FD6 antigens). This could even be supplementary.

Response: This is now provided in Suppl. Figure S2

Critique_ Figure 2: The authors might consider matching the colors in parts b and c.

Response: Modified as suggested.

Critique_ Figure 3: As for Fig. 1, the authors could indicate binding to matched antigens by grouping them.

Response: This is now provided in Suppl. Figure S5

Critique_Text: Line 159 – should have Wibmer et al.

Response: Done (now line 160)

Reviewer #3 The revised manuscript has addressed the cross-reactivity of the immune sera by adding more antigens with matched V1V2 sequences. Overall, the authors have made a good point that the immunization regimen DNA+V1V2-scaffold has elicited higher titers of antibodies conferring complement deposition (C1q binding) and ADCP measured by V2 antigens including cV2 peptides and V1V2 scaffold than other immunization regimens (DNA+ gp120 or SAd7+gp140) involving larger HIV envelope molecules as immunogen. Such antigens could be utilized for detecting V2-specific antibody responses that may confer the above stated functions. The manuscript is quite focused on the elicitation of the V2-specific antibody responses, which would be more balanced by discussing the overall

biological beneficial effect of immunization regimens and presenting data accordingly.

Critique: The authors had provided new data of ADCP and C1q binding functional assay, using cV2 peptide or V1V1 scaffold as antigen. For HIV specific antibody response to confer ADCC/ADCP function, the antibodies have to bind the whole HIV envelope glycoproteins presented on the infected cells. It would be important for the authors to provide data using the whole gp140 trimer as antigen in such assay, and also using infected cells as target cell for ADCC function.

Response: ADCC data have been published and are cited in references 22 and 63 (Hessell et al., *Cell Rep.*, 2019; Powell et al., *J. Virol.* 2020), therefore we do not believe this needs further amplification here.

Response: gp140 trimers were not used for the functional assays for two major reasons: (a) it is improbable that gp140 trimers will distinguish clearly between V2p and V2i antibodies, and (b) there are multiple versions of gp140 trimers and there is controversy as to how much each different molecular iteration actually replicates the trimer on the surface of the infected cell or virus. Consequently, we feel that addition of further ADCC with Env trimers does not add substantially to the message of this paper.

Critique: In reference 62, the difference of ADCC between the anti-Env and anti-V2 immunization regimen is not significant. It may be helpful to assess V2-specific ADCC activity in this assay.

Response: It is important to note that the V2-targeting immunization regimen cited by the reviewer (used in reference 62: Hessell et al., *J. Imm.*, 2021) is different than the one used in our manuscript (and that used in Hessell et al., *Cell Rep.*, 2019 and Powell et al., *J. Virol.*, 2020). It is highly likely that the different immunization regimens are responsible for the different results.

Critique: Adding control V2-specific mAbs in the assays in Fig. 4 could improve the data clarity.

In the Methods section describing the C1q binding assay, the use of the relevant V2-specific mAbs is described as requested by the reviewer. This now has been added to the Methods section on ADCP (lines 550-2).

Critique: Regarding the antigenicity and immunogenicity of C5 region, there has been description in the literature that the exposure of C5 region has been dampened in trimeric gp140 context (*J Virol.* 74(12):5716-25), which may help with the discussion

Response: Text and the relevant citation (#65) have been added as requested (lines 201-3).

Critique: Although it is encouraging that the DNA+V1V2-scaffold regimen has elicited higher V2i antibody response than the other regimens, the authors should discuss the limitation of this regimen.

Response: We note that we had included discussion of limitations of this regimen in the Discussion. We have now added to the existing statements a sentence pointing out that we do not yet know if the responses induced by the regimen we used are protective (425-6). Those experiments are planned.

Critique: Indeed, as reference 62 reported, the ADCC, ADCP, and ADCVI scores of the animals from the anti-V2 regimen were not better than the anti-Env group. The authors stated in lines 421-422 that “In the data presented, only the V2-directed immunization regimen induced Abs with the Fc-mediated effector functions of ADCP and complement activation”. This statement is not consistent with the context of the literature including reference 62. Therefore, this sentence could be revised.

Response: We note again that the V2-targeting regimen in reference 62 is different than the one used here so that comparisons between the data in Hessel, *J. Immunol.*, 2021 and those in our manuscript are not relevant.

Minor points

Critique: There are numerous boxes with bold lines in Figures 1 and 3, which block other areas of the figure and could be removed to improve the clarity of the figures.

Response: This has been corrected.

Critique: Figure 5b please explain what are Dim 2 and Dim 1, and how they have been generated? Either the method part or the figure legend provided sufficient information.

Response: The axes of this figure have been changed to clarify the analysis and this is now noted in the relevant figure legend (line 1026). Similarly, this has been done in Supplementary Figure S7 with the appropriate addition to its figure legend (lines 74-5 of file labeled “Highlighted Supplementary figure legends Re-REVIEW”).